# LEARNING EXPLICIT CREDIT ASSIGNMENT FOR MULTI-AGENT JOINT Q-LEARNING

## ABSTRACT

Multi-agent joint Q-learning based on Centralized Training with Decentralized Execution (CTDE) has become an effective technique for multi-agent cooperation. During centralized training, these methods are essentially addressing the multi-agent credit assignment problem. However, most of the existing methods *implicitly* learn the credit assignment just by ensuring that the joint Q-value satisfies the Bellman optimality equation. In contrast, we formulate an *explicit* credit assignment problem where each agent gives its suggestion about how to weight individual Q-values to explicitly maximize the joint Q-value, besides guaranteeing the Bellman optimality of the joint Q-value. In this way, we can conduct credit assignment among multiple agents and along the time horizon. Theoretically, we give a gradient ascent solution for this problem. Empirically, we instantiate the core idea with deep neural networks and propose Explicit Credit Assignment joint Q-learning (ECAQ) to facilitate multi-agent cooperation in complex problems. Extensive experiments justify that ECAQ achieves interpretable credit assignment and superior performance compared to several advanced baselines.

## 1 INTRODUCTION

Many real-world problems such as robot swarm control can be naturally modeled as cooperative multi-agent systems where each agent can only observe parts of the systems' state and all agents share the same global reward. Recently, the IGM-based multi-agent joint Q-learning has become an effective technique to solve such problems, where IGM (i.e., Individual-Global-Max) means the consistency between individual and joint greedy action selections.

During training, most IGM-based methods (Rashid et al., 2018; Sunehag et al., 2018; Wang et al., 2020b; Yang et al., 2020a;b) are essentially addressing the multi-agent credit assignment problem as pointed out by Yang et al. (2020a); Zhou et al. (2020). Specifically, they try to learn an assignment function $f$ parameterized by $w$ to align the joint Q-value $Q_{total}$ shared by all agents with the individual Q-values $Q_i$ belonging to agent $i$, i.e., $Q_{total} = f(Q_1, ..., Q_N; w)$. Typically, these methods mainly apply temporal difference learning (TD-learning) to extract the parameter $w$, which represents a specific credit assignment. However, TD-learning does not *explicitly* optimize the credit assignment among multiple agents at a given timestep [1], and the existing methods are often called the *implicit* multi-agent credit assignment as mentioned by Zhou et al. (2020); Wang et al. (2020a); Naderializadeh et al. (2020); Li et al. (2021b;a). Besides, the extracted parameter $w$ is usually lack of interpretability in terms of multi-agent credit assignment. Finally, the performance may be poor due to unsuitable credit assignment (Zhou et al., 2020; Yang et al., 2020a).

In contrast, conducting explicit multi-agent credit assignment could distribute the global reward to each agent based on its contribution to the agent group, thus it may substantially facilitate policy optimization and promote learning performance as pointed out by many previous methods (Proper & Tumer, 2012; Tumer & Agogino, 2007; Wang et al., 2020c). More importantly, it can figure out which agent is critical according to the assigned credits, so as to achieve better interpretability, which makes up for the defect that deep neural networks are unexplainable. Inspired by these, we investigate *explicit multi-agent credit assignment* for the IGM-based joint Q-learning methods.

---

[1] In general, TD-learning is considered to explicitly assign the credit along the time horizon, namely, distributing the future credit (i.e., the delayed reward) to previous timesteps.

Our contributions are three-fold. First, we propose a criterion to measure an assignment function so that we can find better assignments by explicitly optimizing this criterion. Specifically, a good assignment function should be helpful for agent cooperation to maximize the reward, so we define the criterion as the maximization of $Q_{total}$ (i.e., the expected long-term cumulative reward) [2]. Second, we introduce an exact solution to optimize the defined criterion. Theoretically, our solution can find the optimal $Q_{total}$ with mild conditions. Empirically, we approximate the core idea with deep neural networks and propose Explicit Credit Assignment joint Q-learning (ECAQ) to facilitate multi-agent cooperation in complex scenarios. Third, we evaluate ECAQ on several challenging tasks. The results demonstrate that ECAQ achieves interpretable credit assignment and superior performance compared to advanced baselines.

## 2 BACKGROUND

**DEC-POMDP.** We consider a *fully cooperative* multi-agent setting that can be formulated as DEC-POMDP (Bernstein et al., 2002). It is formally defined as a tuple $\langle N, S, \boldsymbol{A}, T, R, \boldsymbol{O}, Z, \gamma \rangle$, where $N$ is the number of agents; $S$ is the set of state; $\boldsymbol{A} = A_1 \times ... \times A_N$ represents the set of *joint action*, and $A_i$ is the set of *local action* that agent $i$ can take; $T(s'|s, \boldsymbol{a}) : S \times \boldsymbol{A} \times S \to [0, 1]$ represents the state transition function; $R : S \times \boldsymbol{A} \to \mathbb{R}$ is the reward function; $\boldsymbol{O} = [O_1, ..., O_N]$ is the set of *joint observation* controlled by the observation function $Z : S \times \boldsymbol{A} \to \boldsymbol{O}$; and $\gamma \in [0, 1]$ is the *discount factor*.

In a given state $s$, each agent $i$ generates an action $a_i$ based on its observation $o_i$. The joint action $\boldsymbol{a} = \langle a_i, \boldsymbol{a}_{-i} \rangle$ results in a new state $s'$ (i.e., the state of the next timestep) and a global reward $r$, where $\boldsymbol{a}_{-i}$ is the joint action of teammates of agent $i$. The agent aims at learning a policy $\pi_i(a_i|o_i)$ that can maximize $\mathbb{E}_{o_i \sim Z_i, a_i \sim \pi_i}[G]$ where $G$ is the *discount return* defined as $G = \sum_{t=0}^{H} \gamma^t r^t$ and $H$ is the time horizon. In partially observable scenarios, the action is typically generated based on the entire observation-action *history* $\tau_i$, i.e., $\pi_i(a_i|\tau_i)$, rather than the current observation $o_i$.

**Multi-agent Joint Q-learning.** The multi-agent joint Q-learning is a notable approach to solve DEC-POMDP problems. The idea is to coordinate all agents by the *joint Q-value* $Q_{joint}(\boldsymbol{\tau}, \boldsymbol{a})$ where $\boldsymbol{\tau} = \langle \tau_i, \boldsymbol{\tau}_{-i} \rangle$ is the *joint history* of all agents, then the best joint action can be derived by $\boldsymbol{a}^* = \operatorname{argmax}_{\boldsymbol{a}} Q_{joint}(\boldsymbol{\tau}, \boldsymbol{a})$. In practice, *the true but unknown joint Q-value $Q_{joint}(\boldsymbol{\tau}, \boldsymbol{a})$ is approximated by $Q_{total}(\boldsymbol{\tau}, \boldsymbol{a})$, which in turn is implemented using a deep neural network $Q_{total}(\boldsymbol{\tau}, \boldsymbol{a}; w)$ parameterized by $w$.* Typically, $Q_{total}(\boldsymbol{\tau}, \boldsymbol{a}; w)$ is optimized by minimizing the following TD-loss with temporal difference learning (TD-learning):

$$L(w) = \mathbb{E}_{(\boldsymbol{\tau}, \boldsymbol{a}, r, \boldsymbol{\tau}') \sim D}[(r + \gamma \max_{\boldsymbol{a}'} Q_{total}(\boldsymbol{\tau}', \boldsymbol{a}'; w^-) - Q_{total}(\boldsymbol{\tau}, \boldsymbol{a}; w))^2] \tag{1}$$

where $D$ is the *replay buffer* containing recent experience tuples $(\boldsymbol{\tau}, \boldsymbol{a}, r, \boldsymbol{\tau}')$, and $Q_{total}(\boldsymbol{\tau}, \boldsymbol{a}; w^-)$ is the *target network* whose parameter $w^-$ is periodically updated by copying $w$.

However, vanilla joint Q-learning has some disadvantages. First, the scalability is poor for large-scale agents because it needs to search the whole joint action space to find the optimal one. Second, the agent cannot interact with the environment based on its own information $\tau_i$, since the optimal action also relies on the teammates' information $\boldsymbol{\tau}_{-i}$, namely, $a_i^* \xleftarrow[\text{readout}]{i} \boldsymbol{a}^* = \operatorname{argmax}_{\boldsymbol{a}} Q_{total}^*(\langle \tau_i, \boldsymbol{\tau}_{-i} \rangle, \boldsymbol{a}; w)$.

To remedy these disadvantages, the Centralized Training with Decentralized Execution (CTDE) paradigm is applied (Lowe et al., 2017; Foerster et al., 2018; Sunehag et al., 2018; Rashid et al., 2018). During centralized training, agents are granted access to other agents' information $\boldsymbol{\tau}_{-i}$ (and possibly the global state $s$ if available) to estimate the joint Q-value $Q_{total}(\boldsymbol{\tau}, \boldsymbol{a}; w)$ in a stationary way, while during decentralized execution, the agent makes decision independently based on individual Q-value $a_i = \operatorname{argmax}_{a_i} Q_i(\tau_i, a_i; \theta_i)$ where $\theta_i$ is the policy parameter of agent $i$. In order to achieve effective value-based CTDE, it is critical to ensure the consistency between individual and joint greedy action selections, which induces the Individual-Global-Max (IGM) principle (Son et al., 2019):

$$\langle \operatorname{argmax}_{a_1} Q_1(\tau_1, a_1), ..., \operatorname{argmax}_{a_N} Q_N(\tau_N, a_N) \rangle = \operatorname{argmax}_{\boldsymbol{a}} Q_{joint}(\boldsymbol{\tau}, \boldsymbol{a}) \tag{2}$$

---

[2]We notice that several previous methods (Zhou et al., 2020; Wang et al., 2020e) also take the maximization of $Q_{total}$ as a target or measurement for good credit assignment. The differences are discussed in Section 3.

The IGM principle is very effective to train large-scale agents because it has a linear (rather than exponential) search space for the optimal joint action (compared to the vanilla joint Q-learning). Recently, QTRAN (Son et al., 2019) proposes a sufficient and necessary condition for IGM:

$$\Sigma_{i=1}^N \alpha_i Q_i(\tau_i, a_i) - Q_{joint}(\boldsymbol{\tau}, \boldsymbol{a}) + V_{joint}(\boldsymbol{\tau}) = \begin{cases} 0 & \boldsymbol{a} = [\text{argmax}_{a_i} Q_i(\tau_i, a_i)]_{i=1}^N \\ \geq 0 & \text{otherwise} \end{cases} \quad (3)$$

where $\alpha_i > 0$, and $V_{joint}(\boldsymbol{\tau}) = \max_{\boldsymbol{a}} Q_{joint}(\boldsymbol{\tau}, \boldsymbol{a}) - \Sigma_{i=1}^N \alpha_i \max_{a_i} Q_i(\tau_i, a_i)$ could be interpreted as a baseline function to correct for the discrepancy between the optimal joint Q-value and the weighted summation of the optimal individual Q-values. Note that Equation (2) and (3) are defined under a specific state (equally, under a specific joint history $\boldsymbol{\tau}$) because it is hard to satisfy IGM for all states. Nevertheless, if we could always find a corresponding $\alpha_i$ satisfying Equation (3) for each possible $\boldsymbol{\tau}$, we say the task itself is factorizable (Son et al., 2019).

## 3 RELATED WORK

**IGM-based Credit Assignment.** During centralized training, the IGM-based joint Q-learning methods are essentially addressing the multi-agent credit assignment problem (Yang et al., 2020a; Zhou et al., 2020), namely, aligning the joint Q-value $Q_{total}$ with individual Q-values $Q_i$: $Q_{total}(\boldsymbol{\tau}, \boldsymbol{a}; w) = f([Q_i(\tau_i, a_i; \theta_i)]_{i=1}^N; w)$. The major difference lies in the detailed implementation of the credit assignment function $f$. For example, VDN (Sunehag et al., 2018) proposes a simple additivity assignment function $Q_{total}(\boldsymbol{\tau}, \boldsymbol{a}) = \Sigma_{i=1}^N Q_i(\tau_i, a_i; \theta_i)$, and it works pretty well. QMIX (Rashid et al., 2018) applies a nonlinear assignment function to increase representation expressiveness, but with the constraint of monotonic improvement $\forall i, \frac{\partial Q_{total}(\boldsymbol{\tau}, \boldsymbol{a}; w)}{\partial Q_i(\tau_i, a_i; \theta_i)} \geq 0$ to satisfy the IGM. Recent methods such as WQMIX (Rashid et al., 2020), QTRAN (Son et al., 2019), QPLEX (Wang et al., 2020b), Qatten (Yang et al., 2020b) and QPD (Yang et al., 2020a) propose more sophisticated assignment functions to enhance the representation expressiveness, e.g., the multi-head attention function (Yang et al., 2020b) and the duplex dueling function (Wang et al., 2020b).

**Policy-based Credit Assignment.** A well-known method is COMA (Foerster et al., 2018), which conducts credit assignment by counterfactual baseline. We argue that maximizing $Q_{total}$ is one of the effective ways to learn a good credit assignment function. For example, LICA (Zhou et al., 2020) and DOP (Wang et al., 2020e) take this as the target or measurement of good credit assignment. The key differences between our ECAQ and these methods are two-fold: first, ECAQ is a Q-learning method, while LICA and DOP are actor-critic methods; second, ECAQ explicitly optimizes a credit assignment criterion, while LICA and DOP learn the decomposed credit assignment implicitly.

**Other Methods.** There are other types of multi-agent credit assignment methods (Proper & Tumer, 2012; Tumer & Agogino, 2007; Wang et al., 2020c; Zhang et al., 2020; Zhou et al., 2021) and multi-agent cooperation methods (Nguyen et al., 2018; Zhang et al., 2020; Mahajan et al., 2019; Wang et al., 2020d). However, they are beyond the scope of this paper. Due to space limitation, we provide a brief review for these methods in the Appendix.

## 4 OUR METHOD

Explicit credit assignment is important for achieving better performance and interpretability, but most IGM-based methods only *implicitly* learn the credit assignment function, resulting in non-interpretable assignment and possibly poor performance (Zhou et al., 2020; Yang et al., 2020a). In this section, we investigate *explicit* credit assignment for the IGM-based joint Q-learning. Specifically, Section 4.1 defines the considered problem; Section 4.2 proposes an exact solution to assign credit among multiple agents at a given timestep/state; Section 4.3 approximates the exact solution to handle complex problems; Section 4.4 combines TD-learning with our solution, so the integrated approach can conduct credit assignment among multiple agents and between different timesteps.

### 4.1 PROBLEM FORMULATION

In this paper, the true but unknown joint Q-value $Q_{joint}(\boldsymbol{\tau}, \boldsymbol{a})$ is approximated by $Q_{total}(\boldsymbol{\tau}, \boldsymbol{a})$, which in turn is implemented using a deep neural network $Q_{total}(\boldsymbol{\tau}, \boldsymbol{a}; w)$ parameterized by $w$. Us-

ing these notations, the considered *multi-agent credit assignment function* is formulated as follows:

$$Q_{total}(\boldsymbol{\tau}, \boldsymbol{a}) = \Sigma_{i=1}^N \alpha_i(\tau_i) Q_i(\tau_i, a_i; \theta_i) + b(\boldsymbol{\tau}) \geq Q_{joint}(\boldsymbol{\tau}, \boldsymbol{a}) \tag{4}$$

where $\alpha_i(\tau_i) > 0$ is the weight of $Q_i$, and $b(\boldsymbol{\tau}) := V_{joint}(\boldsymbol{\tau})$. We choose this formulation due to two important reasons. First, it is a sufficient and necessary condition for IGM under some conditions as demonstrated by Equation (3), so it has good fitting ability theoretically (Son et al., 2019; Wang et al., 2020b). Second, it allows us to intuitively interpret the weight $\alpha_i(\tau_i)$ as the importance of $Q_i$, which can be analyzed to understand the concrete credit assignment (please see the experiments).

There are infinite possible assignment solutions satisfying Equation (4). In order to find the best one, we can define a criterion to justify how good an assignment function is, then explicitly optimize such a criterion. We call this kind of methods the *explicit multi-agent credit assignment (MACA)*.

Recall that the ultimate goal of MACA is to boost learning performance, which is measured by the maximization of $Q_{total}$ (i.e., the expected long-term system-level rewards). Thus, we propose an *explicit multi-agent credit assignment criterion* as follows:

$$\{\alpha_i^*(\tau_i), \theta_i^*\} = \underset{\{\alpha_i(\tau_i), \theta_i\}}{\operatorname{argmax}} \ Q_{total}(\boldsymbol{\tau}, \boldsymbol{a}) \tag{5}$$

$$\text{s.t.} \quad Q_{total}(\boldsymbol{\tau}, \boldsymbol{a}) = \Sigma_{i=1}^N \alpha_i(\tau_i) Q_i(\tau_i, a_i; \theta_i) + b(\boldsymbol{\tau}) \ \text{ and } \ \alpha_i(\tau_i) > 0 \ \text{ and } \ \Sigma_{i=1}^N \alpha_i(\tau_i) = 1$$

The additional constraint $\Sigma_{i=1}^N \alpha_i(\tau_i) = 1$ makes sure that $\alpha_i(\tau_i)$ is bounded, so we cannot maximize $Q_{total}$ by simply using an infinite $\alpha_i(\tau_i)$.

## 4.2 AN EXACT SOLUTION FOR MACA

In this section, we introduce an exact solution of Equation (5) under the single-state/timestep setting. This will guide us to design the approximated solution for multi-state problems in Section 4.3. In the following, we omit unnecessary notations (i.e., $\boldsymbol{\tau}$ and $[\tau_i]_{i=1}^N$ due to single state) for simplicity.

Specifically, Equation (5) is a two-objective (i.e., $\boldsymbol{\alpha}^* = [\alpha_i^*]_{i=1}^N$ and $\boldsymbol{\theta}^* = [\theta_i^*]_{i=1}^N$) optimization problem. The Generalized Expectation Maximization (GEM) (Fessler & Hero, 1994) is a popular technique to solve such problems. We adopt this idea and propose a two-stage optimization method. At the *initial weighting stage*, each agent $i$ proposes an initial weighting vector $\boldsymbol{\alpha}_i = [\alpha_i^1, ..., \alpha_i^N]$ that assigns $\alpha_i^l$ weights (i.e., credits) to $Q_l$ to show its preference about how much contribution agent $l$ has made to the agent team. Here, $\boldsymbol{\alpha}_i = [\alpha_i^l]_{l=1}^N$ is agent $i$'s estimation of the optimal $\boldsymbol{\alpha}^* = [\alpha_i^*]_{i=1}^N$. At the *optimization stage*, each agent $i$ iteratively optimizes its weighting vector $\boldsymbol{\alpha}_i$ and its policy parameter $\theta_i$ as follows:

$$\alpha_i^l \leftarrow \alpha_i^l + \beta_1 \frac{1}{N} \Sigma_{j=1}^N (\alpha_j^l - \alpha_i^l) \tag{6}$$

$$\theta_i \leftarrow \Sigma_{j=1}^N \alpha_i^j \theta_j + \beta_2 \nabla_{Q_i} Q_{total}(\boldsymbol{a}) \nabla_{\theta_i} Q_i(a_i; \theta_i) \tag{7}$$

where $\beta_1$ and $\beta_2$ are the learning rates. Equation (6) guarantees that all agents eventually reach consistent weighting vectors based on the difference of them. Equation (7) makes sure that $Q_{total}$ can be maximized by gradient ascent given the weighting vectors. Interlacing the above updates could be seen as a kind of GEM algorithms, and the advantages are two-fold (Blondin & Hale, 2020a;b). First, this "decentralized negotiation" will be more robust to the weighting disagreement among agents compared to a centralized weighting mechanism. Second, it can guarantee that the convergent values $[\alpha_i^{l*}, \theta_i^*]_{i=1}^N$ are optimal under mild assumptions as shown by Proposition 1.

**Proposition 1.** *Under assumption that the individual Q-value functions $[Q_i]_{i=1}^N$ were continuously differentiable and convex, interlacing Equation (6) and (7) enough times will result in convergent $[\alpha_i^{l*}, \theta_i^*]_{i=1}^N$ where $\alpha_i^{l*} = \alpha_j^{l*} = \alpha^*$, and the joint Q-value $Q_{total}(\boldsymbol{a}) = \Sigma_{i=1}^N \alpha_i^* Q_i(a_i; \theta_i^*)$ will converge to the optimal value $Q_{total}^*$.*

*Proof.* First, Olfati-Saber et al. (2007) have proved that updating Equation (6) enough times will make all agents reach the same convergent weighting vector, i.e., $\alpha_i^l = \alpha_j^l$. Second, assuming all $[Q_i]_{i=1}^N$ were continuously differentiable and convex (this is a common assumption, and it is true if the Q-values are linear in "features" as pointed out by Silver et al. (2014)), it is easy to prove that $Q_{total}$ will also be convex given a specific weighting vector; therefore, gradient ascent in Equation

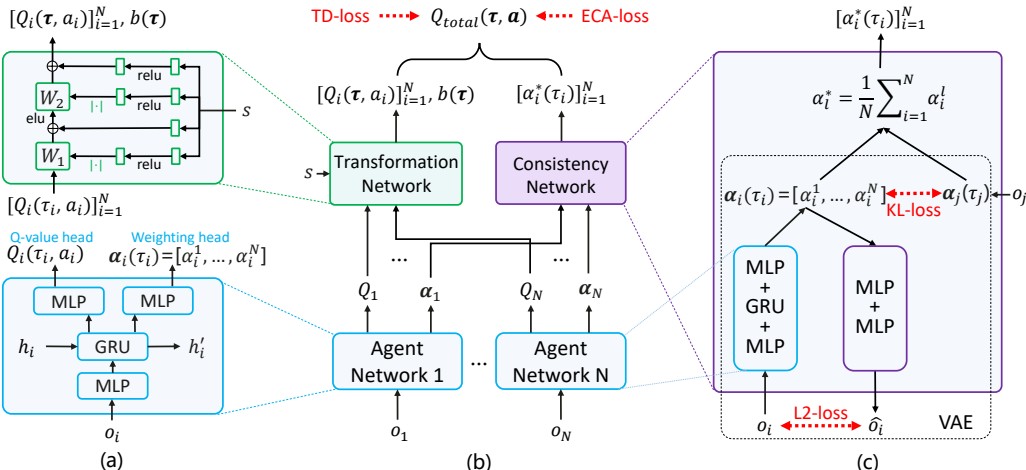

Figure 1: (a) The structures of Agent Network and Transformation Network. (b) The overall ECAQ architecture. (c) The Consistency Network structure.

(7) can always find the maximum point $Q^*_{total}$ with proper learning rate (since the first term does not affect the gradient direction in expectation). Finally, the convergence theory of Expectation Maximization (Wu, 1983; Xu & Jordan, 1996) tells us that interlacing Equation (6) and (7) will not affect the final convergence properties, so we will eventually find the corresponding $[\alpha^*_i, \theta^*_i]^N_{i=1}$. □

### 4.3 APPROXIMATING THE EXACT SOLUTION

The real-world problems often consist of multiple states, but the above solution can hardly learn a set of $[\alpha^*_i, \theta^*_i]^N_{i=1}$ that is optimal for all states because the optimal $[\alpha^*_i, \theta^*_i]^N_{i=1}$ is state-dependent. In general, a policy with strong fitting ability can alleviate this problem, so we approximate the exact solution using deep neural networks (DNN) as the function approximator.

**Overall Design.** Our DNN-based method is called ECAQ, which consists of three sub-networks as shown in Figure 1. The Agent Network first applies a GRU to encode the local observation $o_i$ into the history $\tau_i$. Then, it uses the Q-value head and the weighting head to generate the individual Q-value $Q_i(\tau_i, a_i; \theta_i)$ and the weighting vector $\boldsymbol{\alpha}_i(\tau_i; \theta_i) = [\alpha^1_i, ..., \alpha^N_i]$, respectively. The weighting head adopts a Softmax activation function to guarantee $\Sigma^N_{l=1} \alpha^l_i = 1$.

The Consistency Network is proposed to *optimize the weighting vector*. It takes as input the original weighting vectors from all agents, and optimizes them to the same converged value $[\alpha^*_i(\tau_i; \theta_i)]^N_{i=1}$ (i.e., mimicking the effect of Equation (6)). The details will be introduced in the following section.

The Transformation Network is proposed to increase ECAQ's representation expressiveness. As previous methods, it uses the joint history $\boldsymbol{\tau}$ (or the state $s$ if available) to generate a two-layer hypernetwork (Ha et al., 2017), then transforms individual Q-value $[Q_i(\tau_i, a_i; \theta_i)]^N_{i=1}$ to $[Q_i(\boldsymbol{\tau}, a_i; \theta_i, w)]^N_{i=1}$ using this hypernetwork. Besides, it also generates a baseline function $b(\boldsymbol{\tau}; w)$.

Finally, we form the joint Q-value as $Q_{total}(\boldsymbol{\tau}, \boldsymbol{a}; w) = \Sigma^N_{i=1} \alpha^*_i(\tau_i; \theta_i) Q_i(\boldsymbol{\tau}, a_i; \theta_i, w) + b(\boldsymbol{\tau}; w)$. Afterwards, we can *optimize the policy parameter* to the converged value $[\theta^*_i]^N_{i=1}$ (i.e., mimicking the effect of Equation (7)). The details will be introduced in the following section.

**Optimizing the Weighting Vector.** As mentioned before, the Consistency Network is proposed to optimize the weighting vectors $[\boldsymbol{\alpha}_i]^N_{i=1}$ to the same converged value. The main difficulty is how to apply DNN to achieve this goal. Here, we adopt the variational inference technique (Mao et al., 2020b) with the following key idea: assuming that the optimal weighting vector is $\boldsymbol{\alpha}^* = [\alpha^*_i]^N_{i=1}$, but it is unknown; each agent $i$ would like to infer $\boldsymbol{\alpha}^*$ based on $o_i$ by $p(\boldsymbol{\alpha}^*|o_i)$; if all agents' weighting vectors $[\boldsymbol{\alpha}_i]^N_{i=1}$ could really converge to the same $\boldsymbol{\alpha}^*$, we would achieve our goal.

In practice, directly computing $p(\boldsymbol{\alpha}^*|o_i) = \frac{p(o_i|\boldsymbol{\alpha}^*)p(\boldsymbol{\alpha}^*)}{\int p(o_i|\boldsymbol{\alpha}^*)p(\boldsymbol{\alpha}^*)d\boldsymbol{\alpha}^*}$ is quite difficult, so we approximate $p(\boldsymbol{\alpha}^*|o_i)$ using another tractable distribution $q(\boldsymbol{\alpha}^*|o_i)$ by minimizing the KL-divergence between

them, namely, $\min KL(q(\boldsymbol{\alpha}^*|o_i)||p(\boldsymbol{\alpha}^*|o_i))$, which equals to:

$$\max \mathbb{E}_{q(\boldsymbol{\alpha}^*|o_i)} \log p(o_i|\boldsymbol{\alpha}^*) - KL(q(\boldsymbol{\alpha}^*|o_i)||p(\boldsymbol{\alpha}^*)) \tag{8}$$

Equation (8) can be modeled by a variational autoencoder (VAE), which is the main part of the Consistency Network. The encoder of this VAE learns a mapping $q(\boldsymbol{\alpha}_i|o_i; \theta_i)$ from $o_i$ to $\boldsymbol{\alpha}_i$, and the decoder learns a mapping $p(\widehat{o_i}|\boldsymbol{\alpha}_i; \theta_i)$ from $\boldsymbol{\alpha}_i$ back to $\widehat{o_i}$. The loss function to train this VAE is:

$$L_i^{vae}(\theta_i) = L2(o_i, \widehat{o_i}; \theta_i) + KL(q(\boldsymbol{\alpha}_i|o_i; \theta_i)||p(\boldsymbol{\alpha}^*)) \tag{9}$$

where the first term represents the reconstruction error of observations/states, and minimizing this error makes sure that the weighting vector $\boldsymbol{\alpha}_i(\tau_i; \theta_i)$ is state-dependent so as to better handle real-world problems consisting of multiple states; the second term ensures that the learned distribution $q(\boldsymbol{\alpha}_i|o_i; \theta_i)$ is similar to the true prior distribution $p(\boldsymbol{\alpha}^*)$. However, the true prior $p(\boldsymbol{\alpha}^*)$ in Equation (9) is unknown. One could assume that $p(\boldsymbol{\alpha}^*)$ follows a unit Gaussian distribution as previous methods, but it cannot be true for all states. In practice, we find that other agents' weighting vector $q(\boldsymbol{\alpha}_j|o_j; \theta_j)$ is a good surrogate for $p(\boldsymbol{\alpha}^*)$, namely, we approximate Equation (9) by:

$$L_i^{vae}(\theta_i) \approx L2(o_i, \widehat{o_i}; \theta_i) + \frac{1}{N}\Sigma_{j=1}^N KL(q(\boldsymbol{\alpha}_i|o_i; \theta_i)||q(\boldsymbol{\alpha}_j|o_j; \theta_j)) \tag{10}$$

Provably, Proposition 2 shows that Equation (10) has the same effect as Equation (6) under single-state setting. Intuitively, this approximation punishes any pair of agents $\langle i, j \rangle$ with inconsistent weighting vectors, namely, with a large $KL(q(\boldsymbol{\alpha}_i|o_i; \theta_i)||q(\boldsymbol{\alpha}_j|o_j; \theta_j))$. Therefore, it is helpful for converging to the same weighting vector. In practice, the above optimization cannot be iterated infinitely, so we make sure that the weighting vectors are eventually consistent by averaging them $\alpha_l^* \approx \frac{1}{N}\Sigma_{i=1}^N \alpha_i^l$.

**Proposition 2.** *Under single-state setting, Equation (10) has the same effect as Equation (6).*

*Proof.* For single-state, some common assumptions hold: 1) there is no need to recover observation, so the first term of Equation (10) is removed; 2) $q(\boldsymbol{\alpha}_i)$ and $q(\boldsymbol{\alpha}_j)$ are Gaussians with equal standard deviation; therefore, minimizing $\frac{1}{N}\Sigma_{j=1}^N KL(q_i||q_j) = \frac{1}{N}\Sigma_{j=1}^N \log \frac{\sigma_j}{\sigma_i} + \frac{\sigma_i^2 + (\mu_i - \mu_j)^2}{2\sigma_j^2} - \frac{1}{2} = \frac{1}{N}\Sigma_{j=1}^N (\mu_i(\alpha_i) - \mu_j(\alpha_j))^2$ has the same effect as minimizing $\frac{1}{N}\Sigma_{j=1}^N (\alpha_j - \alpha_i)$ in Equation (6). $\square$

**Optimizing the Policy Parameter.** In practice, we share policy parameters among agents as the previous methods (e.g., VDN, QMIX and QTRAN). Besides accelerating convergence, the special advantage is that the first term of Equation (7) can be simplified as $\Sigma_{j=1}^N \alpha_i^j \theta_j = \theta_i$, therefore Equation (7) can be rewritten as:

$$\theta_i \leftarrow \theta_i + \beta_2 \nabla_{Q_i} Q_{total}(\boldsymbol{\tau}, \boldsymbol{a}; w) \nabla_{\theta_i} Q_i(\tau_i, a_i; \theta_i) \tag{11}$$

The loss function of Equation (11) is $L_i^{eca}(\theta_i) = -Q_{total}$. We call it Explicit Credit Assignment loss (ECA-loss) because it explicitly maximizes $Q_{total}$ (i.e., the criterion of good credit assignment).

## 4.4 PUTTING IT ALL TOGETHER

The exact solution proposed in Section 4.2 is summarized by Equation (6) and (7). In Section 4.3, ECAQ adopts the VAE-loss (i.e., Equation (10)) and ECA-loss (i.e., Equation (11)) to approximate Equation (6) and (7), respectively. Nevertheless, VAE-loss and ECA-loss are mainly used to optimize the credit assignment among multiple agents at a given timestep/state. For the sequential decision-making problems consisting of multiple timesteps/states, it is also critical to do credit assignment along the time horizon (i.e., assigning the delayed reward to previous timesteps). Therefore, ECAQ also minimizes the following TD-loss:

$$L^{td}(w) = \mathbb{E}_{(\boldsymbol{\tau}, \boldsymbol{a}, r, \boldsymbol{\tau}') \sim D}[(r + \gamma \max_{\boldsymbol{a}'} Q_{total}(\boldsymbol{\tau}', \boldsymbol{a}'; w^-) - Q_{total}(\boldsymbol{\tau}, \boldsymbol{a}; w))^2] \tag{12}$$

It ensures the Bellman optimality of $Q_{total}$, namely, $Q_{total}$ can approximate the true but unknown $Q_{joint}$ very closely. Putting it all together, the total loss to train ECAQ is:

$$L(w, \theta_i) = L^{td}(w) + \eta(\Sigma_{i=1}^N L_i^{vae}(\theta_i) + \Sigma_{i=1}^N L_i^{eca}(\theta_i)) \tag{13}$$

where $\eta$ is the hyperparameter to balance the credit assignment among multiple agents and between different timesteps. The detailed training algorithm is provided in the Appendix.

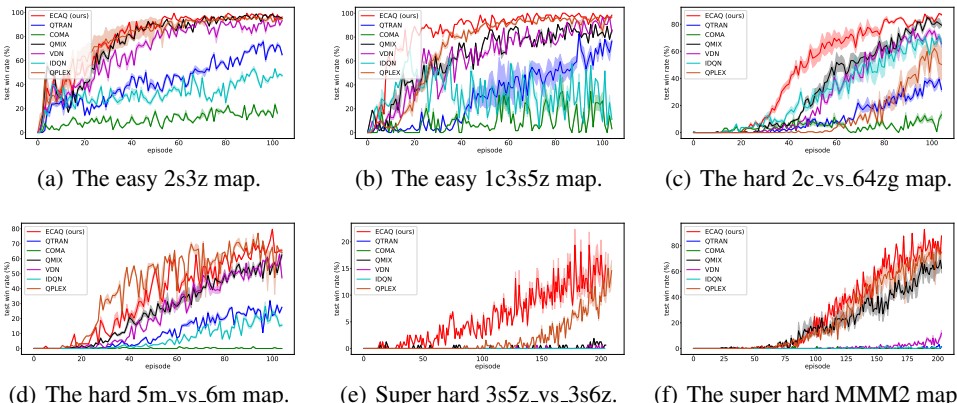

(a) The easy 2s3z map.    (b) The easy 1c3s5z map.    (c) The hard 2c_vs_64zg map.

(d) The hard 5m_vs_6m map.    (e) Super hard 3s5z_vs_3s6z.    (f) The super hard MMM2 map.

Figure 2: The test win rate on different StarCraft II maps. ECAQ achieves the best performance on four maps (i.e., the easy 1c3s5z, the hard 2c_vs_64zg, and the super hard 3s5z_vs_3s6z and MMM2), and performs as good as QPLEX on two maps (i.e., the easy 2s3z and the hard 5m_vs_6m).

## 5 EXPERIMENT

**Environment.** To guarantee a fair comparison, the decentralized StarCraft II micromanagement problem (Samvelyan et al., 2019) is used since it is usually considered as the official testbed for the IGM-based methods. Besides, previous works show that StarCraft II is suitable for credit assignment study (Zhou et al., 2020; Yang et al., 2020a; Wang et al., 2020e; Foerster et al., 2018). Specifically, six maps with different configurations (e.g., easy, hard, and super hard settings; homogeneous and heterogeneous agents) are used to guarantee that ECAQ does not overfit to one specific scenario.

We also evaluate ECAQ on the cooperative navigation problem (Lowe et al., 2017; Mordatch & Abbeel, 2018), which is a simple yet popular multi-agent environment. Specifically, there are $N$ agents and $N$ landmarks on a 10-by-10 2D plane. The agents are controlled by our methods, and they try to cover all landmarks. The observation is the relative positions and velocities of other agents and landmarks. The action is the velocity of agents. The reward is the negative distance of any agent to each landmark. We test three scenarios where $N = 4$, 6 and 10, respectively.

**Baseline.** Since we study the explicit credit assignment for the IGM-based joint Q-learning, here we mainly compare with these IGM-based methods. Specifically, we consider the most relevant VDN (Sunehag et al., 2018), QMIX (Rashid et al., 2018) and QTRAN (Son et al., 2019). We also adopt COMA (Foerster et al., 2018) and IDQN (Tampuu et al., 2017). COMA applies a counterfactual baseline to do credit assignment; in contrast, there is no credit assignment in IDQN. The advanced methods like QPLEX (Wang et al., 2020b), WQMIX (Rashid et al., 2020), LICA (Zhou et al., 2020), DOP (Wang et al., 2020e) and Qatten (Yang et al., 2020b) are also reported.

**Implementation.** Since most baselines are officially provided in the PyMARL framework [3], we also use this framework to implement ECAQ and to conduct the experiments. We do not modify any of the default configurations/hyperparameters of PyMARL to guarantee a fair comparison. For the special hyperparameter $\eta$ of ECAQ, we decrease it from 1.0 to 0.05 gradually as training goes on.

### 5.1 MAIN RESULT

**Result of StarCraft II.** The average test win rates of five independent runs are shown in Figure 2. As can be observed, ECAQ achieves the best performance on four maps (i.e., the easy 1c3s5z, the hard 2c_vs_64zg, and the super hard 3s5z_vs_3s6z and MMM2), and performs as good as QPLEX on two maps (i.e., the easy 2s3z and the hard 5m_vs_6m). Notably, the performance of COMA is unstable: it works well in some scenarios but it is even worse than IDQN in other scenarios. These results demonstrate that a good credit assignment is very necessary for consistent multi-agent cooperation. In order to compare ECAQ with advanced methods, we show the results on hard and super hard maps

---

[3]https://github.com/oxwhirl/pymarl.

Table 1: The test win rate on the hard and super hard StarCraft II maps.

| Map | Qatten | QPLEX | WQMIX | LICA | DOP | ECAQ (ours) |
|---|---|---|---|---|---|---|
| hard 2c_vs_64zg | 66 | 55 | 67 | 68 | **84** | **85** |
| hard 5m_vs_6m | **72** | **70** | 60 | 60 | 63 | **73** |
| super hard 3s5z_vs_3s6z | **17** | **12** | 6 | 0 | 0 | **15** |
| super hard MMM2 | **79** | **72** | 23 | **84** | 50 | **80** |
| average test win rate | 58.5 | 52.25 | 39.0 | 53.0 | 49.25 | **63.25** |

in Table 1. As can be seen, ECAQ's performance is better than or as good as many advanced methods (e.g., Qatten, QPLEX and WQMIX) in specific maps, while its average performance is the best, although ECAQ does not involve advanced DNN architectures like duplex dueling or multi-head attention critic. This is because ECAQ can directly learn a good credit assignment that maximizes the long-term rewards, which is highly positive for the test win rate. For example, in the super hard 3s5z_vs_3s6z scenario, ECAQ assigns a high credit to ally's Zealots because the learned policy is that ally's Zealots hold enemy's Zealots and attack enemy's Stalkers at the same time.

**Result of Cooperative Navigation.** The average rewards of ten independent runs are shown in Figure 3. It can be observed that VDN outperforms QMIX in cooperative navigation, which is in contrast to the results of StarCraft II where QMIX is better than VDN. This highlights the relationship among method performance, method complexity and task complexity: complex methods do not always get better performance in simple tasks. Nevertheless, as can be seen, ECAQ obtains more rewards than other baselines in most scenarios. It seems that the complexity of the evaluated tasks does not influence ECAQ too much.

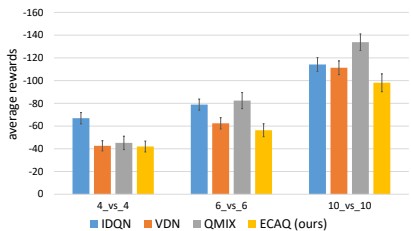

Figure 3: The average rewards in cooperative navigation. Lower bar is better.

## 5.2 FURTHER ANALYSIS

**Ablation Study.** The average test win rate is shown in Figure 4, where TD, ECA and VAE represent the three loss functions defined before (namely, TD+ECA+VAE stands for ECAQ). Surprisingly, it can be observed that neither TD+ECA nor TD+VAE performs well, and they are somehow worse than simply applying a single TD-loss. The reason may be that 1) TD+ECA cannot reach consistent weighting vectors due to the lack of VAE-loss, so the optimization of ECA-loss may be incorrect; 2) TD+VAE does not optimize the credit assignment at all due to the lack of ECA-loss. Therefore, both TD+ECA and TD+VAE deteriorate the solution. These results assert a conclusion that both VAE-loss and ECA-loss are necessary for ECAQ's good performance. We also find the same conclusion in the matrix games, and the details are shown in the Appendix.

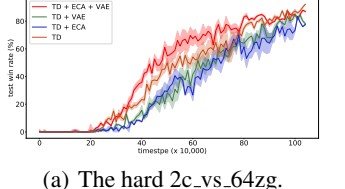
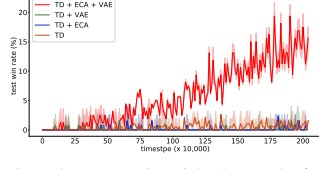
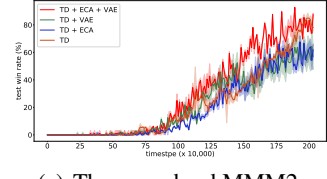

(a) The hard 2c_vs_64zg.      (b) The super hard 3s5z_vs_3s6z.      (c) The super hard MMM2.

Figure 4: The ablation results for (super) hard scenarios. Other scenarios are shown in the Appendix.

**Credit Assignment Study.** To make a clear analysis, we should make the environment as simple as possible, so we adopt the matrix games in this study. Firstly, we use the asymmetric monotonic game $G$ as shown in Table 2(a). There are two agents in $G$ and each agent $i$ has three actions $a_{iA}$, $a_{iB}$, and $a_{iC}$. The asymmetricity means that different agents have different impact on the payoff. Specifically, the column agent (i.e., agent 2) has larger impact compared to the row agent (i.e., agent 1) in game $G$, since the payoff changes by two units in column while by one unit in row. We train ECAQ through a full exploration conducted over 20,000 steps as QTRAN (Son et al., 2019). We

Table 2: The asymmetric monotonic game and the converged results on the game.

(a) Payoff of the matrix game $G$.

| $a_1$ \ $a_2$ | $a_{2A}$ | $a_{2B}$ | $a_{2C}$ |
|---|---|---|---|
| $a_{1A}$ | 7 | 5 | 3 |
| $a_{1B}$ | 6 | 4 | 2 |
| $a_{1C}$ | 5 | 3 | 1 |

(b) $Q_1$, $Q_2$ and $Q_{total}$ learned by ECAQ under setting #1.

| $Q_1$ \ $Q_2$ | $\mathbf{4.828}(a_{2A})$ | $2.107(a_{2B})$ | $-0.611(a_{2C})$ |
|---|---|---|---|
| $\mathbf{6.639}(a_{1A})$ | **7.004** | 5.005 | 3.007 |
| $2.854(a_{1B})$ | 6.000 | 4.001 | 2.003 |
| $-0.927(a_{1C})$ | 4.997 | 2.998 | 1.000 |

(c) The converged values. ECAQ has learned the optimal policy (i.e., $a_{1A}$ and $a_{2A}$).

| Setting | $\alpha_1^*$ | $\alpha_2^*$ | $Q_1^*$ | $Q_2^*$ | $Q_{total}^*$ |
|---|---|---|---|---|---|
| #1: $G$ | 0.26523 | 0.73477 | $6.639(\mathbf{a_{1A}})$ | $4.828(\mathbf{a_{2A}})$ | 7.004 |
| #2: $G^T$ | 0.72562 | 0.27438 | $4.755(\mathbf{a_{1A}})$ | $6.435(\mathbf{a_{2A}})$ | 6.982 |
| #3: $G * 10$ | 5.83e-08 | 1.00e+00 | $55.900(\mathbf{a_{1A}})$ | $33.236(\mathbf{a_{2A}})$ | 63.738 |

Table 3: The symmetric non-monotonic game and the converged results on the game.

(a) Payoff of the matrix game.

| $a_1$ \ $a_2$ | $a_{2A}$ | $a_{2B}$ | $a_{2C}$ |
|---|---|---|---|
| $a_{1A}$ | 8 | -12 | -12 |
| $a_{1B}$ | -12 | 0 | 0 |
| $a_{1C}$ | -12 | 0 | 0 |

(b) The converged values in different settings.

| Setting | $\alpha_1^*$ | $\alpha_2^*$ | $Q_{total}^*$ |
|---|---|---|---|
| #1: full exploration | 0.55601 | 0.44399 | -3.568 |
| #2: high probability for $a_{1A}$ | 1.35e-07 | 9.99e-01 | 0.666 |
| #3: high probability for $a_{2A}$ | 9.99e-01 | 9.09e-08 | 0.151 |

check how different payoff settings will affect the credit assignment. As shown in Table 2(c), ECAQ assigns a large weight to agent 2 in game $G$ (i.e., $\alpha_2^* \approx 0.735$ in setting #1) because agent 2 has larger impact on the payoff. In contrast, when we transpose the payoff of $G$, denoted by $G^T$, the weight of agent 2 will be small (i.e., $\alpha_2^* \approx 0.274$ in setting #2) since agent 2 is less influential on the payoff in $G^T$. We further increase the payoff of $G$ by ten times, denoted by $G * 10$, and the weight of agent 2 changes to a very large value (i.e., $\alpha_2^* \approx 1.0$ in setting #3). This is because the payoff changes by twenty units (rather than the original two units) in column, and the absolute impact of agent 2 becomes much larger. These analyses have demonstrated the good credit assignment ability of ECAQ. Consequently, ECAQ can easily find the optimal decentralized policy (i.e., $a_{1A}$ and $a_{2A}$) in all games as observed from Table 2(c). Finally, comparing Table 2(b) with 2(a), it can be seen that ECAQ fits the payoff matrix very accurately, which asserts the good fitting ability of ECAQ.

Secondly, we use the symmetric non-monotonic game as shown in Table 3(a). This game is proposed by QTRAN (Son et al., 2019). We check how different exploration settings will affect the credit assignment. As shown in Table 3(b), the full exploration (i.e., setting #1 as QTRAN) will result in almost random credit assignment values (i.e., $\alpha_i^* \approx 0.5$). This is expected because the payoff matrix is symmetric and the exploration is full, so there is no difference between the two agents. In contrast, when we make one agent more stable (e.g., raising the probability of $a_{1A}$ in setting #2), ECAQ will assign a large weight to focus on the other agent (e.g., $\alpha_2^* \approx 1.0$ in setting #2). The reason is that the other agent is more random, and it has greater impact on the obtained reward. The results of setting #3 are similar to these of setting #2, and both settings can find the optimal policy as shown in the Appendix. Overall, these analyses have demonstrated the good credit assignment ability of ECAQ.

## 6  CONCLUSION

Multi-agent credit assignment has long been a fundamental issue for multi-agent cooperation. This paper presented the *explicit multi-agent credit assignment* for the IGM-based joint Q-learning, which not only ensures the Bellman optimality of $Q_{total}$ to do credit assignment along the time horizon, but also optimizes a criterion to do credit assignment among different agents explicitly. We instantiate this idea with deep neural networks and propose ECAQ to facilitate multi-agent cooperation in more realistic scenarios. Extensive experiments justify the superior performance of ECAQ. Furthermore, the detailed analyses show that ECAQ has really learned interpretable credit assignment values. To our best knowledge, the explicit credit assignment is complementary yet novel to the existing IGM-based studies. We believe that it is basic for building effective learning-based multi-agent systems.

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

## A   THE TRAINING ALGORITHM

ECAQ adopts the Centralized Training with Decentralized Execution (CTDE) paradigm. The training algorithm for ECAQ is shown in Algorithm 1.

---

**Algorithm 1:** Training Algorithm for ECAQ

---

**Input:** Randomly initialized $\theta_i$ and $w$ for the policy networks (i.e., individual Q-value functions $Q_i(\tau_i, a_i; \theta_i)$) and the mixing critic (i.e., the joint Q-value function $Q_{total}(\boldsymbol{\tau}, \boldsymbol{a}; w)$).
**Output:** Converged individual Q-value $Q_i^*(\tau_i, a_i; \theta_i^*)$ for future decentralized execution.

1   $Q_{total}(\boldsymbol{\tau}, \boldsymbol{a}; w^-) \leftarrow Q_{total}(\boldsymbol{\tau}, \boldsymbol{a}; w)$;
2   **while** *not terminated* **do**
3     The agents interact with the environment based on $Q_i(\tau_i, a_i; \theta_i)$, and put the experience tuples $(s, \boldsymbol{\tau}, \boldsymbol{a}, r, \boldsymbol{\tau}')$ into replay buffer $D$;
4     Sample batch size of tuples $(s, \boldsymbol{\tau}, \boldsymbol{a}, r, \boldsymbol{\tau}')$ from replay buffer $D$;
5     **for** *each tuple $(s, \boldsymbol{\tau}, \boldsymbol{a}, r, \boldsymbol{\tau}')$* **do**
6       Each agent $i$ extracts its observation $o_i$;
7       The Agent Network generates individual Q-values $Q_i(\tau_i, a_i; \theta_i)$ and weighting vector $\boldsymbol{\alpha}_i(\tau_i; \theta_i) = [\alpha_i^1, ..., \alpha_i^N]$ based on $o_i$, and further generates the decoded observation $\hat{o}_i$ based on $\boldsymbol{\alpha}_i(\tau_i)$;
8       Calculate the VAE-loss as $L_i^{vae}(\theta_i) \approx L2(o_i, \hat{o}_i; \theta_i) + \frac{1}{N}\Sigma_{j=1}^N KL(q(\boldsymbol{\alpha}_i|o_i; \theta_i)||q(\boldsymbol{\alpha}_j|o_j; \theta_j))$;
9       The Consistency Network takes as input $[\boldsymbol{\alpha}_i]_{i=1}^N$ and outputs the converged $[\boldsymbol{\alpha}_i^*(\tau_i; \theta_i)]_{i=1}^N$;
10      The Transformation Network takes as input $[Q_i(\tau_i, a_i; \theta_i)]_{i=1}^N$ and state $s$, then generates $[Q_i(\boldsymbol{\tau}, a_i; \theta_i, w)]_{i=1}^N$ and $b(\boldsymbol{\tau}; w)$;
11      Calculate the joint Q-value as $Q_{total}(\boldsymbol{\tau}, \boldsymbol{a}; w) = \Sigma_{i=1}^N \alpha_i^*(\tau_i; \theta_i)Q_i(\boldsymbol{\tau}, a_i; \theta_i, w) + b(\boldsymbol{\tau}; w)$;
12      Calculate the ECA-loss as $L^{eca}(w) = -Q_{total}(\boldsymbol{\tau}, \boldsymbol{a}; w)$;
13      Calculate the TD-loss as $L^{td}(w) = (r + \gamma \max_{\boldsymbol{a}'} Q_{total}(\boldsymbol{\tau}', \boldsymbol{a}'; w^-) - Q_{total}(\boldsymbol{\tau}, \boldsymbol{a}; w))^2$;
14      Calculate the total loss as $L(w, \theta_i) = L^{td}(w) + u\Sigma_{i=1}^N L_i^{vae}(\theta_i) + v\Sigma_{i=1}^N L_i^{eca}(\theta_i)$;
15      Train the network parameters $[\theta_i]_{i=1}^N$ and $w$ by back-propagation based on the total loss;
16     **end**
17     # In practice, the above for iteration is processed in a mini-batch manner;
18     **if** *at target update interval* **then**
19       Update the target mixing critic by $Q_{total}(\boldsymbol{\tau}, \boldsymbol{a}; w^-) \leftarrow Q_{total}(\boldsymbol{\tau}, \boldsymbol{a}; w)$;
20     **end**
21 **end**

---

## B   CREDIT ASSIGNMENT ANALYSIS

### B.1   CREDIT ASSIGNMENT ANALYSIS FOR COOPERATIVE NAVIGATION

We analyze the training behaviors of the weighting value (i.e., credit assignment value) $\alpha_i$ ($i = 1, 2, 3$) using the 3_vs_3 cooperative navigation. We first analyze the dynamics of $\alpha_1 = \frac{1}{N}\Sigma_{i=1}^N \alpha_i^1$ and agent $i$'s estimation $\alpha_i^1$ ($i = 1, 2, 3$), which are shown by the four solid lines in Figure 5(a). As shown by the black, green and blue lines, although the values are very different at the beginning of training, different $\alpha_i^1$ ($i = 1, 2, 3$) tend to be consistent as training goes on. We then analyze whether the converged $\alpha_i$ ($i = 1, 2, 3$) is meaningful. Figure 5(b) shows the learned policies by the red arrows, and the credit assignment values (which can be estimated from Figure 5(a)) are as follows : $\alpha_1 \approx 0.47$ since agent A1 is far from the landmark and its policy has the largest influence on the reward; $\alpha_2 \approx 0.35$ and $\alpha_3 \approx 0.18$ since agent A2 and A3 are near the landmark and their policies have little influence on the reward. The results are consistent with these shown in the main paper: the most influential agent usually corresponds to the largest assignment value. These analyses indicate that the assignment values $\alpha_i$ ($i = 1, 2, 3$) are meaningful to identify critical agents.

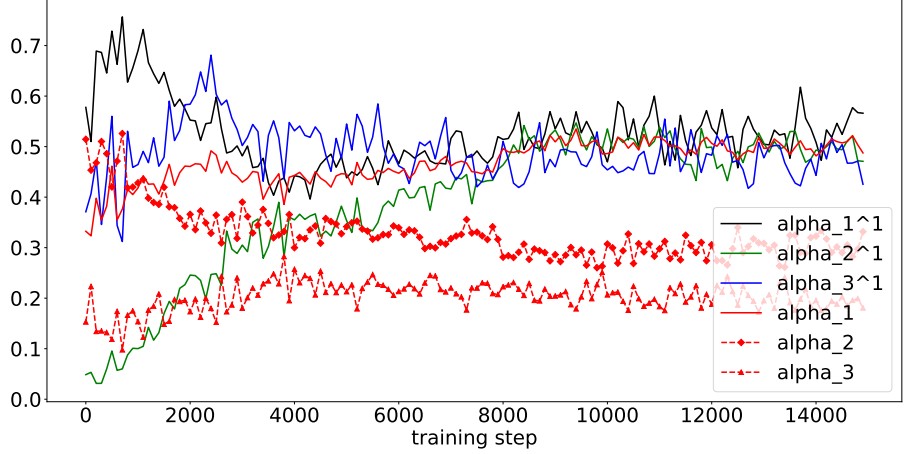

(a) The training dynamic of $\alpha_i$ and agent $i$'s estimation $\alpha_i^1$ of $\alpha_1$.

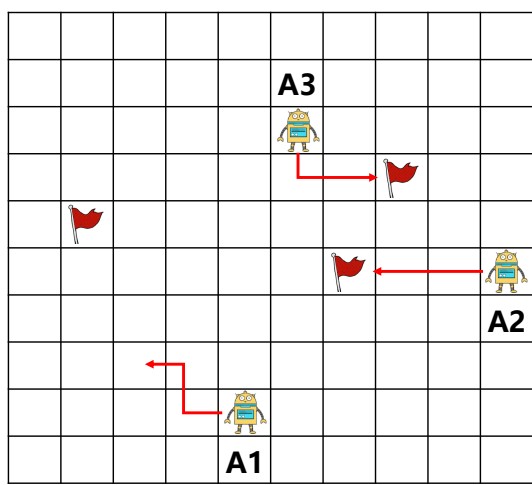

(b) $\alpha_i$ is meaningful for MACA. A$i$ represents agent $i$.

Figure 5: The credit assignment analysis using the 3_vs_3 cooperative navigation task.

Table 4: The payoff matrix of the one-step game and the converged results on the game.

(a) Payoff of the matrix game.

| $a_1 \diagdown a_2$ | $\boldsymbol{a_{2A}}$ | $a_{2B}$ | $a_{2C}$ |
|---|---|---|---|
| $\boldsymbol{a_{1A}}$ | **8** | -12 | -12 |
| $a_{1B}$ | -12 | 0 | 0 |
| $a_{1C}$ | -12 | 0 | 0 |

(b) $Q_1$, $Q_2$ and $Q_{total}$ learned by ECAQ under setting #3.

| $Q_1 \diagdown Q_2$ | **-0.014**$(a_{2A})$ | -2.497$(a_{2B})$ | -3.062$(a_{2C})$ |
|---|---|---|---|
| **1.674**$(a_{1A})$ | **0.1509** | 0.1505 | 0.1505 |
| -5.407$(a_{1B})$ | -6.9395 | -6.9395 | -6.9300 |
| -6.487$(a_{1C})$ | -8.0108 | -8.0108 | -8.0109 |

(c) The converged values for different settings. Note that both #2 and #3 can find the optimal policy.

| Setting | $\alpha_1^*$ | $\alpha_2^*$ | $Q_1^*$ | $Q_2^*$ | $Q_{total}^*$ |
|---|---|---|---|---|---|
| #1: fully exploration | 0.56 | 0.44 | 0.0403$(a_{1C})$ | 0.6296$(a_{2C})$ | -3.568 |
| #2: high probability for $a_{1A}$ | 1.35e-07 | 9.99e-01 | 0.4702$(\boldsymbol{a_{1A}})$ | 1.7616$(\boldsymbol{a_{2A}})$ | 0.666 |
| #3: high probability for $a_{2A}$ | 9.99e-01 | 9.09e-08 | 1.6739$(\boldsymbol{a_{1A}})$ | -0.0140$(\boldsymbol{a_{2A}})$ | 0.151 |
| #4: training only by TD-loss | 9.99e-01 | 2.58e-06 | 0.4889$(a_{1A})$ | -0.8592$(a_{2B})$ | -0.695 |

### B.2 CREDIT ASSIGNMENT ANALYSIS FOR MATRIX GAME

We also adopt a matrix game (Son et al., 2019) to check the credit assignment ability of ECAQ. As shown in Table 4(a), there are two agents and each agent $i$ has three actions $a_{iA}$, $a_{iB}$ and $a_{iC}$. We check how different settings will affect the credit assignment. As shown in Table 4(c), the fully exploration (i.e., setting #1 as QTRAN (Son et al., 2019)) will result in almost random credit assignment value (i.e., $\alpha_i^* \approx 0.5$). This is expected because the payoff matrix is symmetric and the exploration is full, so there is no difference between the two agents. In contrast, when we make one agent more stable (e.g., raising the probability of $a_{1A}$ in setting #2), ECAQ will assign large weights to focus on the other agent (e.g., $\alpha_2^* \approx 1.0$ in setting #2). The reason is that the other agent is more random, and it has greater impact on the obtained reward. The results of setting #3 are similar to these of setting #2, and both settings can find the optimal decentralized policy (i.e., $a_{1A}$ and $a_{2A}$) as observed. Comparing the results shown in Table 4(b) with these shown in (Son et al., 2019; Wang et al., 2020b), it can be seen that ECAQ fits the optimal payoff more accurately than VDN, QMIX and Qatten. Overall, these analyses have demonstrated the good credit assignment ability and fitting ability of ECAQ.

## C   MORE ABLATION STUDY

We conduct ablation study on the easy maps (e.g., 2s3z), finding that there is no significant performance difference for different ablation models as shown by Figure 6(a). The reason is that the maps are too easy, and all methods can get good results. Therefore, we focus on doing ablation study on the hard and super hard maps. As shown in the main paper, neither TD+ECA nor TD+VAE performs well, and they are somehow worse than simply applying a single TD-loss in the hard 2c_vs_64zg, super hard 3s5z_vs_3s6z and super hard MMM2 maps. However, for the hard 5m_vs_6m scenarios, TD+ECA, TD+VAE and TD have almost similar performance as shown by Figure 6(b), but they are worse than TD+ECA+VAC (i.e., ECAQ). In the future, we will do more ablation study on different maps and draw more detailed conclusions when the computing resources are available.

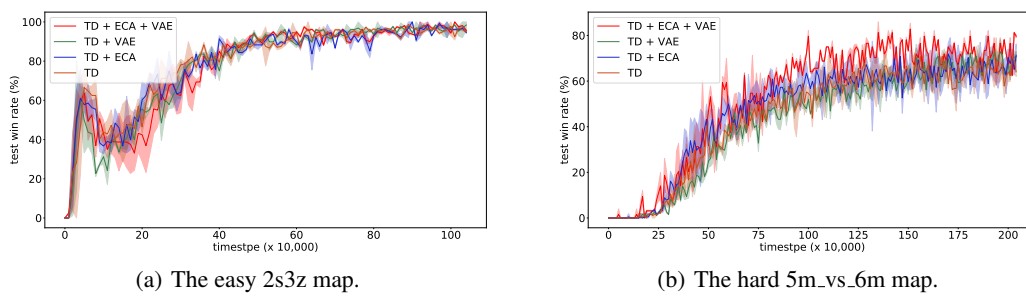

(a) The easy 2s3z map.                                    (b) The hard 5m_vs_6m map.

Figure 6: The ablation results on StarCraft II.

We also conduct ablation study on the matrix game shown in Table 4(a). The payoff matrix is symmetric, so there will be no difference between the two agents under full exploration. Therefore, we raise the probability of $a_{2A}$ (i.e., setting #3 in Table 4(c)), and ECAQ can find the optimal decentralized policy (i.e., $a_{1A}$ and $a_{2A}$) as observed. In contrast, if we train ECAQ only by TD-loss with the same exploration (i.e., setting #4), agent 2 can just find a non-optimal policy $a_{2B}$, which asserts the necessity of both VAE-loss and ECA-loss.

## D   RELATED WORK

### D.1   MULTI-AGENT CREDIT ASSIGNMENT APPROACH

In this section, we provide a brief review about the multi-agent credit assignment approaches that are closely related to the proposed ECAQ.

**Explicit Credit Assignment.** In general, the explicit methods attribute agent contributions that are at least locally optimal (Kinnear, 1994). A notable approach is to assess an action by calculating *difference reward* against a certain reward baseline (Tumer & Agogino, 2007; Agogino & Turner, 2005; Proper & Tumer, 2012). The key idea is that the true contribution of agent $i$ can be approximated by the difference between rewards induced by $a$ and $[a_c, a_{-i}]$, where $a_c$ is a counterfactual action (i.e., not the true action agent $i$ has taken). For example, Agogino & Turner (2005) and Tumer & Agogino (2007) adopt specific $a_c$ to calculate the difference reward or CLEAN reward to do credit assignment. COMA (Foerster et al., 2018) and SQDDPG (Wang et al., 2020c) extends this idea from reward difference to Q-value difference, and calculates the counterfactual baseline or Shapley Q-value to do credit assignment. Specifically, COMA (Foerster et al., 2018) uses a centralized critic to estimate the *counterfactual advantage* of an action. SQDDPG (Wang et al., 2020c) applies the Shapley-value framework (Shapley, 1988) to do credit assignment based on an agent's *marginal contribution* as it is sequentially added to possible agent groups. However, the *explicit* mentioned here is different from the *explicit* mentioned by ECAQ.

**The IGM-based Approach.** The representative methods are VDN (Sunehag et al., 2018), QMIX (Rashid et al., 2018), QTRAN (Son et al., 2019), Qatten (Yang et al., 2020b), QPLEX (Wang et al., 2020b), etc. We have reviewed these methods in the main paper. They are often called the *implicit* credit assignment methods as noted by Zhou et al. (2020). In contrast, ECAQ applies *explicit* credit assignment training signal to optimize the IGM-based approaches.

**The Attribution Approach.** The key idea is that split the final reward into two kinds of rewards: self-reward and attributed reward; and the attributed reward is assumed to be redistributed by other agents based on the agent's contribution. Methods following this idea are (Nguyen et al., 2018; Zhang et al., 2020; Mao et al., 2020a), and they can also be seen as explicit credit assignment.

**Other Approach.** There are many other multi-agent credit assignment methods that can be hardly classified clearly, for example, the implicit credit assignment (Zhou et al., 2020), the social reward credit assignment (Mataric, 1994) and others (Mannion et al., 2017; Grefenstette, 1995).

### D.2 MULTI-AGENT COOPERATION APPROACH

Please note that we aim at giving a complementary thought for the multi-agent credit assignment problem rather than beating all methods (with hyperparameter tuning), so we implement ECAQ based on the basic QMIX instead of the advanced approaches like Qatten (Yang et al., 2020b) and QPLEX (Wang et al., 2020b). Therefore, we do not intend to give a detailed review for all recent deep MARL methods. Nevertheless, there are many topics to facilitate multi-agent cooperations, for example, the role-based methods (Mahajan et al., 2019), the coordinative exploration methods (Wang et al., 2020d), the graph-based methods (Blondin & Hale, 2020a;b; Mao et al., 2020b) and so on. However, these methods are beyond the scope of this paper.

## E  IMPLEMENTATION DETAILS OF ECAQ

As mentioned in the main paper, we use the up-to-date PyMARL framework [4] to conduct the experiments. We do not modify any of the default configurations of PyMARL to guarantee a fair comparison. We do not tune the hyperparameters of ECAQ too much: the weights of VAE-loss and ECA-loss are directly set equal, and they decrease from $1.0$ to $0.05$ gradually as training goes on; all of the other hyperparameters (e.g., exploration and activation functions) keep the same as these of PyMARL. The code is available on the submission system. Therefore, the main experimental results can be easily reproduced. In addition, we train our methods on the 64-core CPU machine with a total memory of 128G. The training time is 4 hours to 10 hours for different StarCraft II maps. This is not too computing heavy for MARL applications.

---

[4]https://github.com/oxwhirl/pymarl. The code licensed under the Apache License v2.0, so we can use it freely for research purpose.

# F    LIMITATIONS OF ECAQ

ECAQ is implemented based on deep neural networks, so it faces the "black box problem" where the behaviors of individual agents may not be interpretable from the perspective of human. Fortunately, ECAQ adopts an explicit manner to do multi-agent credit assignment, and the learned weighting vectors are interpretable to some extend.

ECAQ focuses on the fully cooperative setting, so the goal is set as the maximization of the long-term shared global reward. As a result, the multi-agent credit assignment (MACA) may raise ethical issues when the optimal joint actions require sacrificing certain agents. Certainly, we can define the "fair-criterion" for MACA, then optimize this criterion to guarantee fairness between agents, and this is our future work.

ECAQ is an IGM-based method. The main limitation is that it cannot fit the non-monotonic payoff perfectly, as shown in Table 4(b). The reason is that both the Transformation Network and the weighting vector $\alpha_i$ are nonnegative. However, as mentioned before, we aim at giving a complementary thought for the multi-agent credit assignment problem rather than beating all methods, so we implement ECAQ by an easy-to-explain joint Q-value function. If we implemented ECAQ based on more advanced approaches like Qatten, QTRAN++ or QPLEX, it will be much easier to fit the non-monotonic payoff.

