# OpenReview forum: "Learning Explicit Credit Assignment for Multi-agent Joint Q-learning"
_ICLR.cc/2022/Conference — ICLR 2022 Submitted_

### Official Review · Reviewer_trcZ · 2021-10-25

**Correctness:** 1
**Technical Novelty And Significance:** 3
**Empirical Novelty And Significance:** 2
**Recommendation:** 3
**Confidence:** 4

**Main Review:**

My score is completely based on the fact that the proposed method does not optimize credit assignment in any way for the MARL objective. If the authors can convince me otherwise, I would be glad to change my review.

The paper gives the impression that it aims to do optimal credit assignment. The abstract says "we formulate an explicit credit assignment problem where each agent gives its suggestion about how to weight individual Q-values to explicitly maximize the joint Q-value,". This is then formally written as the optimization problem in equation (5), where the weighs $\alpha$ are supposed to be optimized so as to maximize $Q_{\text{total}}$. But then, in the proposed solution in equations (6) and (7), the $\alpha$ update does not depend on any notion of performance. The $\theta_i$ update depends on $Q$, but the $\alpha$ updates are not affected by $Q$ at all. Even if the initial set of $\alpha^l_i$ converge to some consistent values, they are not the "optimal" weights since the update rule doesn't explicitly optimize any objective. This discrepancy between the intention of the method and what it actually does is also shown later in writing, where the paper says "the Consistency Network is proposed to optimize the weighting vectors $[\alpha_i]_{i=1}^N$ to the same converged value". Creating a process for a set of values to converge doesn't mean that the process optimizes anything. A possible counterargument is that the updates (6) and (7) causes the "optimality" to be contained in the $Q_i$ weights $\theta_i$ rather than in $\alpha_i$. But that is not at all explicit.

Another flaw is that the so-called 'explicit multi-agent credit assignment criterion" in equation (5) is problematic. Essentially, part of it says: given a fixed functional form of $Q_{\text{total}}$, which is some function of $\theta_i$ via the $Q_i$ but not a function of the Markov dynamics (since that is absent in the expressions), find $\theta_i$ that maximizes $Q_{\text{total}}$. So one can just choose $\theta_i = \pm \infty$, depending on signs. Perhaps the authors meant to use an RL objective, like the expected cumulative return, but that is not what equation (5) says.

I do like the overall intention to do explicit multi-agent credit assignment. I believe that some adjustments to the approach will make this work suitable for another submission.

Minor points:
- grammar, e.g. "Besides, the extracted parameter w is usually lack of interpretability"
- "First, this “decentralized negotiation” will be more robust to the weighting disagreement among agents compared to a centralized weighting mechanism." In a centralized weighting scheme, there is no such thing as "disagreement", so what is being compared? The decentralized one has non-zero disagreement.
- equation (9), is the "L2" supposed to be the $l_2$ error norm? How is this equivalent to maximizing the first term in equation (8)? Why wasn't the standard VAE reconstruction loss used?
- The colorbars in the legend in Figure 2 are too thin, I can't distinguish them and match them to the curves easily even at 200% magnification.

**Summary Of The Paper:**

This paper tries to tackle the multi-agent credit assignment problem by an explicit method. Specifically, the method is based on individual-global-max criteria for value decomposition methods for MARL, whereby the joint $Q$-value depends on a weighted sum of individual $Q_i$. The method interprets the individual weight $\alpha_i$ as the "credit" (i.e, "importance") of agent $i$, then lets each agent $i$ have its own guess $\alpha^l_i$ of the importance of another agent $l$, then proposes an update rule to achieve consistency among all agents' guesses. This is first formulated for the stateless case, and then extended to the temporal setting with function approximation. Experiments were conducted on the StarCraft micromanagement benchmark and cooperative navigation to show performance versus baselines, and in matrix games to show the dependence of weights $\alpha$ on payoff asymmetry.

**Summary Of The Review:**

This paper purports to optimize credit assignment for the MARL objective but the method does not do so.

---

> ### Author Response · Authors · 2021-11-12
> **Thanks for your valuable and insightful comments.**
>
> Thanks for your valuable and insightful comments. Below we will address your concerns in detail.
>
> **Question**: In the proposed solution in equations (6) and (7), the alpha update does not depend on any notion of performance. Even if the initial set of alpha converge to some consistent values, they are not the "optimal" weights since the update rule doesn't explicitly optimize any objective. A possible counterargument is that the updates (6) and (7) causes the "optimality" to be contained in the Q_i weights theta_i rather than in alpha_i. But that is not at all explicit.
> **Answer**: Thanks for your suggestion and effort. We are sorry to make you confused. As you can see from Figure 1, the Q_total is function of both Q_i and alpha_i in our practical implementation. It means that is alpha_i is also updated by both equations (6) and (7), i.e., TD-loss and ECA-loss. Therefore, the update about alpha_i will causes the "optimality" of Q_total. We thank the reviewer for this question.
>
>
>
> **Question**: To find that maximizes, one can just choose theta_i=infinite, depending on signs. Perhaps the authors meant to use an RL objective, like the expected cumulative return, but that is not what equation (5) says.
> **Answer**: In our paper, equation (5) is just to demonstrate that we want use the maximization of Q_total as our explicit multi-agent credit assignment criterion. This criterion is certainly combined with our RL objective in practice, as shown by equation (13). Moreover, the weight of TD-loss keeps 1.0 during the whole training procedure, but the weight of ECA-loss decreases from 1.0 to 0.05 as mentioned in the experiments, which means that we take the RL objective as the main objective besides the credit assignment objective. We thank the reviewer for this question.
>
> **Question**: Minor points.
> **Answer**: Thanks for your suggestion and effort. We will revise the grammar errors and make the paper easier to follow in the next version.

---

> > ### Comment · Reviewer_trcZ · 2021-11-18
> > **I am not convinced that the process of updating $\alpha_i$ to some convergence value can be interpreted as optimizing them for multi-agent credit assignment**
> >
> > Yes, I see from Figure 1 that the $Q_{\text{total}}$ is a function of both $Q_i$ and $\alpha_i$. Yes, I see that $\alpha_i$ is updated by minimizing the TD-loss $L^{\text{td}}(w)$. By the widely-accepted interpretation of the TD-loss, which the authors acknowledge in Section 4.4 before equation (12), this means the $\alpha_i$ is optimized for temporal credit assignment, and here, the particular choice for the TD target is the 1-step return in equation (12). This has nothing to do with multi-agent credit assignment. One may try to say that minimizing this TD-loss is optimizing the joint policy performance, and since $Q_{\text{total}}$ is an explicit function of $\alpha_i$, hence minimizing the TD-loss is doing "explicit credit assignment". But if that argument holds, then one might as well say that the first layer of the mixing network in QMIX [Rashid et al. 2018] are "explicit credit assignment weights", since that layer acts on the individual $Q_i$ and QMIX also minimizes the same TD-loss. In sum, either the proposed method in this paper has the same level of "explicitness" as QMIX, or else both the proposed method and QMIX do not perform credit assignment. I favor the latter conclusion, since the former conclusion would mean that QMIX already suffices for explicit credit assignment and there is nothing left for us to do.
> >
> > As for the ECA-loss (equation 11), one can see that the gradient $\nabla_{\theta_i} L_i^{\text{eca}}(\theta_i)$ only flows through the left branch of the agent network in Figure 1, not the right branch that outputs the weights $\alpha_i$ (the shared layers are interpreted as feature embeddings). So, just like equation (7), this optimization affects the parameters of individual $Q_i$ but does nothing to the $\alpha_i$.
> >
> > Combining these two conclusions, I retain my original conclusion that the process of updating $\alpha_i$ to some convergence value cannot be interpreted as optimizing them for multi-agent credit assignment.
> >
> > More comments:
> > - When the authors wrote in their rebuttal "alpha_i is also updated by both equations (6) and (7), i.e., TD-loss and ECA-loss", I believe the authors actually meant to say that equation (6) is the analogue of the VAE loss $L^{\text{vae}}_i$, not the TD-loss, since the authors themselves say in Section 4.4. that "The exact solution proposed in Section 4.2 is summarized by Equation (6) and (7). In Section 4.3, ECAQ adopts the VAE-loss (i.e., Equation (10)) and ECA-loss (i.e., Equation (11)) to approximate Equation (6) and (7), respectively."
> > - The notation for parameters is confusing. It appears that the parameters $w$ of $Q_{\text{total}}$ include both the $\theta_i$ of the individual $Q_i$ and the weights $\alpha_i$, so that $w = [\theta_1,\dotsc,\theta_n,\alpha_1,\dotsc,\alpha_n]$ (maybe it includes even more, since there is a hypernetwork). But later in Section 4.3, it turns out that $\alpha_i$ itself is a function of $\theta_i$, since the authors use the notation $\alpha_i(\tau_i; \theta_i)$. This is made worse by the fact that some gradients, e.g. for the ECA-loss, are written with respect to the whole $\theta_i$, but the author's architecture diagram shows that the gradients don't affect the part of $\theta_i$ that produce the $\alpha_i$. This makes it really confusing. Overloading of notation for ease of presentation should never come at the expense of clarity.

---

> > > ### Author Response · Authors · 2021-11-20
> > > **Thanks for your valuable comments.**
> > >
> > > **Question1:** In sum, either the proposed method in this paper has the same level of "explicitness" as QMIX, or else both the proposed method and QMIX do not perform credit assignment. I favor the latter conclusion, since the former conclusion would mean that QMIX already suffices for explicit credit assignment and there is nothing left for us to do.
> > > **Answer1:** We thank the reviewer for this question. However, we (and many other researchers may also) do not agree with the reviewer because many existing studies have pointed out that QMIX (and other IGM-based methods) are essentially addressing the multi-agent credit assignment problem during centralized training, but in an implicit manner. Please see the Related Work section for details. In contrast, our method is explicit credit assignment in the sence that 1) we can intuitively interpret the weight $\alpha_i$ as the importance of $Q_i$ and 2) we have explicit loss function (besides TD-loss) to optimized $\alpha_i$.
> > >
> > > **Question2:** As for the ECA-loss (equation 11), one can see that the gradient of ECA-loss only flows through the left branch of the agent network in Figure 1, not the right branch that outputs the weights $\alpha_i$ (the shared layers are interpreted as feature embeddings). So, just like equation (7), this optimization affects the parameters of individual $Q_i$ but does nothing to the $\alpha_i$.
> > > **Answer2:** As the reviewer has seen that we use a notation $\alpha_i(\tau_i; \theta_i)$, which means that $\alpha_i$ is a function parameterized by the policy parameter $\theta_i$ (we have clarified this in the Background section). Since the parameters $\theta_i$ of $Q_i(\tau_i, a_i; \theta_i)$ and $\alpha_i(\tau_i; \theta_i)$ are mostly shared (This can be seen from the Agent Network in Figure 1), the ECA-loss  (equation 11) can definitely optimize $\alpha_i$ (as well as individual $Q_i$).  We thank the reviewer for this valuable question.
> > >
> > > **Question3:** Combining these two conclusions, I retain my original conclusion that the process of updating to some convergence value cannot be interpreted as optimizing them for multi-agent credit assignment.
> > > **Answer3:** Yes, **only** updating $\alpha_i$ to some convergence value (i.e., only VAE-loss) cannot optimize the multi-agent credit assignment. That is why we combine the VAE-loss and ECA-loss to optimize the multi-agent credit assignment among different agents, and also combine the TD-loss to optimize the multi-agent credit assignment along the time horizon.
> > >
> > > **Question4:** I believe the authors actually meant to say that equation (6) is the analogue of the VAE loss $L^{vae}_i$, not the TD-loss.
> > > **Answer4:** Yes, the reviewer understand it correctly.
> > >
> > > **Question5:** It appears that the parameters $w$ of $Q_{total}$ include both the $\theta_i$ of the individual $Q_i$ and the weights $\alpha_i$, so that $w=[\theta_1,...,\theta_n,\alpha_1,...,\alpha_n]$ (maybe it includes even more, since there is a hypernetwork). But later in Section 4.3, it turns out that $\alpha_i$ itself is a function of $\theta_i$, since the authors use the notation $\alpha_i(\tau_i;\theta_i)$. This is made worse by the fact that some gradients, e.g. for the ECA-loss, are written with respect to the whole $\theta_i$, but the author's architecture diagram shows that the gradients don't affect the part of $\theta_i$ that produce the $\alpha_i$. This makes it really confusing.
> > > **Answer5:** We thank the reviewer for the effort to try to understand our paper. However, there are some misunderstandings, especially for the notations that may further affect the whole understanding of our paper.
> > > - The parameters $w$ of $Q_{total}$ do not include $\theta_i$ and $\alpha_i$ at all. As we have clarified in the Background section that ''$Q_{total}(\tau,\textbf{a})$ is implemented using a deep neural network $Q_{total}(\tau,\textbf{a}; w)$ parameterized by $w$'', it means that $w$ is the parameter of $Q_{total}(\tau,\textbf{a}; w)$, and please note that the parameter in turn is generated by a hypernetwork as most IGM-based methods do. This is a big mismatch point between the reviewer and our paper, and we hope our answer has made this more clear.
> > > - For the notation $\theta_i$,  we have also clarified in the Background section that ''individual Q-value $Q_i(\tau_i, a_i; \theta_i)$ where $\theta_i$ is the policy parameter of agent $i$''. In practice, it generates both the individual Q-value $Q_i(\tau_i, a_i; \theta_i)$ and the weighting value $\alpha_i(\tau_i;\theta_i)$ as shown by the Agent Network in Figure 1. This is related to the **Question2** proposed by the reviewer. We hope our answer can make the notation more clear.
> > >
> > > **In summary, we are really thankful for these valuable questions proposed by Reviewer trcZ. We are happy to answer more questions to make the paper more easy to understand. We deeply appreciate it if Reviewer trcZ could revalue the technical innovation and overall contributions of the paper. Thanks.**

---

> > > > ### Comment · Reviewer_trcZ · 2021-11-27
> > > > **1) Author has agreed that the formulation of method in Section 4.2 does not optimize credit assignment, which is a fatal flaw; 2) The implementation of the method with function approximation is motivated by Section 4.2, but the practical implementation has significant deviation from the meaning of Section 4.2 and is insufficiently justified.**
> > > >
> > > > The author claims that previous value-decomposition methods perform "implicit" credit assignment (interestingly, none of the original papers that proposed VDN, QMIX and QTRAN interpreted their method in this manner). The concept of "implicit credit assignment" appears to be obfuscation when one really thinks about it. One cannot hide behind the word "implicit": either we can point to something and say, "that's where credit assignment happens", or it is meaningless to call the act of finding an optimal Q-function "implicit credit assignment". But, suppose for the sake of argument that we accept the claim. Let us ask: where does the credit assignment happen in QMIX? Well, it can only be the mixing network that maps the collection of individual $Q_i$ to a global $Q$. The mixing network is quite an explicit object. It has weights for each individual $Q_i$ that can be interpreted by further analysis to see the relative contribution of each $Q_i$ to the total $Q$. So, if the author wants to claim that QMIX performs credit assignment simply by minimizing the TD-loss on a certain combination of individual $Q_i$, then it appears that QMIX has the same level of explicitness as this paper's approach.
> > > >
> > > > But, I doubt that the author wants to claim that minimizing the TD-loss improves credit assignment. From the paper, the authors claim that multi-agent credit assignment is achieved by the combination of equation (6) and (7). Equation (6) is a consensus update, and the authors have agreed with my original review that this update has nothing to do with optimizing credit assignment. Equation (7) increases $Q_{\text{total}}$, not by using any reward from the environment, but simply by updating the function parameters to increase the output value. As the author says, Equation (7) is not a TD-loss. So, the impression from reading Section 4.2, which has nothing to do with rewards or TD-learning at all, is that minimizing a TD-loss is unnecessary for credit assignment. Therefore, there is no evidence in the paper to justify the claim that one can use a "TD-loss to optimize the multi-agent credit assignment along the time horizon."
> > > >
> > > > Now, let's look at equation (7). The author agrees that equation (7) has nothing to do with $\alpha_i$. The author uses equation (7) to motivate equation (11), which is similar to equation (7) except that $Q_i$ is now parameterized by $\theta_i$. Equation (11) is used for practical implementation. Now, it turns out that in the author's practical implementation, both $Q_i$ and $\alpha_i$ share a subset of learnable parameters $\theta_i$, so equation (11) now affects $\alpha_i$. So, as far as $\alpha_i$ is concerned (and the whole focus of this paper is really on $\alpha_i$ since they are the credit assignment weights), the meaning of equation (11) differs from equation (7). In sum: 1) the iteration of equations (6-7) is supposed to be the paper's groundwork for credit assignment, but they do not optimize credit assignment weights $\alpha_i$ for any objective (author has agreed on this point); 2) equations (6-7) have significant differences from the practical implementation; 3) therefore the practical implementation is insufficiently justified.
> > > >
> > > > I understand the point about parameters $w$ of $Q_{\text{total}}$. That was a side comment about clarity and doesn't impinge on the main issues raised above.
> > > >
> > > > Re-reading the author's first-round replies, I see the sentence "Moreover, the weight of TD-loss keeps 1.0 during the whole training procedure, but the weight of ECA-loss decreases from 1.0 to 0.05 as mentioned in the experiments." That raises a big concern. If a mechanism for credit assignment is being learned correctly during training, then that mechanism should continue running indefinitely throughout training and that mechanism should always be helping to improve performance. The fact that the author had to slowly turn off that mechanism suggests that it does not continue to be beneficial to performance. It is hard to see why a properly learned credit assignment would hurt performance, unless, that mechanism isn't really doing credit assignment.

---

> > > > > ### Author Response · Authors · 2021-11-28
> > > > > **Thanks for your valuable and insightful comments.**
> > > > >
> > > > > **Q1: The author claims that previous value-decomposition methods perform "implicit" credit assignment (interestingly, none of the original papers that proposed VDN, QMIX and QTRAN interpreted their method in this manner). The concept of "implicit credit assignment" appears to be obfuscation when one really thinks about it**
> > > > > In fact, we just follow many previous studies to call QMIX-like methods "implicit" credit assignment. This has been mentioned in the paper.  Yes, we do notice that the original papers that proposed VDN, QMIX and QTRAN did not interpreted their method in this manner.
> > > > >
> > > > >
> > > > > **Q2: Therefore, there is no evidence in the paper to justify the claim that one can use a "TD-loss to optimize the multi-agent credit assignment along the time horizon."**
> > > > > As mentioned above, we just follow many previous studies (which have been accepted by NeurIPS, ICML, ICLR etc.) to call QMIX-like methods "implicit" credit assignment. Following this fact, QMIX is only trained by TD-loss, so the TD-loss can optimize (some kinds of) the multi-agent credit assignment? We are sorry to use "along the time horizon", this is indeed not justified by the paper. We will improve the writing in the next version.
> > > > >
> > > > >
> > > > > **Q3: therefore the practical implementation is insufficiently justified.**
> > > > > Thanks for the question. We have realised that our paper writting is not so clear. In Equation (5), we writed that credit assignment weights as $\alpha_i(\tau_i)$ and individual Q-value as $Q_i(\tau_i, a_i;\theta_i)$, so we have been meaning to represent that $\alpha_i(\tau_i)$ and $Q_i(\tau_i, a_i;\theta_i)$ share some parameters (since both condition on $\tau_i$) as our practical implementation. We will improve the writing in the next version.
> > > > >
> > > > >
> > > > > **Q4: If a mechanism for credit assignment is being learned correctly during training, then that mechanism should continue running indefinitely throughout training and that mechanism should always be helping to improve performance.**
> > > > > We did run the mechanism indefinitely throughout training, but just decreasing the weight of the mechanism. This is easy to understand: at the beginning, all agents are randomly initilized, they do not have good sense about credit assignment, so the weight of credit assignment loss should be large; as the training goes on, the agents have learned some skills to assign suitable credit assignment, so there is no need to use a large weight (since TD-learning is the most important for RL).
> > > > >
> > > > >
> > > > > **Overall, we thank the reviewer for the valuable and insightful comments. We will improve our paper in the next version.**

---

> > > > > > ### Comment · Reviewer_trcZ · 2021-11-30
> > > > > > **Acknowledgement of author's reply.**
> > > > > >
> > > > > > I acknowledge the authors for their efforts in defending their paper. I have tried to provide concise comments and supporting arguments to show the need to improve the method and improve its justification. I hope that the authors see this discussion in a positive light.

---

> > > > > > > ### Author Response · Authors · 2021-11-30
> > > > > > > **Thanks.**
> > > > > > >
> > > > > > > We thank Reviewer trcZ for the efforts and valuable time to improve our paper. We do see this discussion in a positive light. Thanks again.

---

### Official Review · Reviewer_wZYm · 2021-11-01

**Correctness:** 2
**Technical Novelty And Significance:** 1
**Empirical Novelty And Significance:** 2
**Recommendation:** 3
**Confidence:** 3

**Main Review:**

I believe this paper aims to tackle a fundamental problem for MARL, i.e., the credit assignment problem. However, I find that most discussions and presentations in this paper are rather vague, which is very hard to understand for me.

I have many questions on the main content:

- Eq.(4) is a VDN-like value factorization. Why do the citations refer to QTRAN and QPLEX?

- In Eq.(4), why $Q_{total}\geq Q_{joint}$?

- The ground-truth objective is $Q_{joint}$. Why do we need to maximize $Q_{total}$ in Eq.(5)?

- I do not understand the constraint $\sum_{i=1}^N \alpha_i=1$. When all agents have full observations, the value of $\alpha$ seems to be a one-hot vector (i.e., a greedy maximization).

- I do not understand why we need to introduce GEM. A trivial maximization solution of Eq.(5) seems to be $Q_i\equiv\infty$.

- Given that $\alpha$ represents an explicit credit assignment, why these values are generated by a VAE? Proposition 2 definitely ignores the reconstruction loss.

**Summary Of The Paper:**

This paper aims to establish an explicit multi-agent credit assignment approach with interpretability.

The proposed method, Explicit Credit Assignment joint Q-learning (ECAQ), is an end-to-end solution to the designed problem formulation. It can leverage the advanced deep-learning-based implementation of MARL based on CTDE.

**Summary Of The Review:**

I vote for rejection since I cannot capture the main ideas of the proposed method from the current version. Most presented arguments do not make sense for me.

---

> ### Author Response · Authors · 2021-11-12
> **Thanks for your valuable and insightful comments.**
>
> Thanks for your valuable and insightful comments. Below we will address your concerns in detail.
>
> **Question**: Eq.(4) is a VDN-like value factorization. Why do the citations refer to QTRAN and QPLEX?
> **Answer**: Since QTRAN proved that Eq.(4) is a sufficient and necessary condition for IGM under some conditions as demonstrated by Equation (3), and QPLEX demonstrate that it has good fitting ability theoretically.
>
> **Question**: In Eq.(4), why Q_total ≥ Q_joint?
> **Answer**: Please note that this is our formulation. We want to make sure that Q_total ≥ Q_joint so that Eq.(4) is a sufficient and necessary condition for IGM under some conditions.
>
> **Question**: The ground-truth objective is Q_joint. Why do we need to maximize Q_total in Eq.(5)?
> **Answer**: We want to make sure that Q_total ≥ Q_joint so that Eq.(4) and Eq.(5) is a sufficient and necessary condition for IGM under some conditions.
>
> **Question**: I do not understand the constraint sigma_alpha=1. When all agents have full observations, the value of alpha seems to be a one-hot vector (i.e., a greedy maximization).
> **Answer**: Thanks for your suggestion and effort. But the reviewer may be wrong, since one-hot vector for alpha cannot produce the maximization Q_total.
>
> **Question**: I do not understand why we need to introduce GEM. A trivial maximization solution of Eq.(5) seems to be Qi≡∞.
> **Answer**: Infinite Qi will results in Infinite Q_total (due to monotonic relation between Qi and Q_total). It is easy to see that Infinite Q_total is not the right one for any discounted MDP.
>
> **Question**: Given that alpha represents an explicit credit assignment, why these values are generated by a VAE? Proposition 2 definitely ignores the reconstruction loss.
> **Answer**: we generate alpha because we want to **learn** the explicit credit assignment. Proposition 2 is proved under single-state setting, where there is no need to recover observation, so the reconstruction loss in Equation (10) is removed.

---

> > ### Comment · Reviewer_wZYm · 2021-11-24
> > **I am not convinced and keep my vote for rejection.**
> >
> > The response of the authors does not address any of my concerns.
> >
> > **Q1: Since QTRAN  proved that Eq.(4) is a sufficient and necessary condition for IGM under some conditions as demonstrated by Equation (3), and QPLEX demonstrate that it has good fitting ability theoretically.**
> >
> > The IGM condition in QTRAN is maintained by an additional loss to the TD-learning. Do you include that loss function into your method? It requires training a fully centralized value network.
> >
> > In addition, I think the authors do not understand what QPLEX did. The theoretical property of QPLEX is given by its special mixing network rather than the VDN-like sum-form factorization.
> >
> > **Q2&Q3: We want to make sure that Q_total ≥ Q_joint so that Eq.(4) is a sufficient and necessary condition for IGM under some conditions.**
> >
> > I cannot see what you want to "*make sure*" through this formulation. $Q\_{joint}$ is defined as the ground-truth unknown value. How is related to the IGM property? It seems that the authors either do not understand or try to confuse the definition of IGM.
> >
> > **Q4: But the reviewer may be wrong, since one-hot vector for alpha cannot produce the maximization Q_total.**
> >
> > Could you please expand more on this argument? A trivial maximization is setting the value to $1$ at the maximum-value $Q_i$, which leads to an one-hot vector.
> >
> > I know you consider other loss functions to train $\alpha_i$. My concern is that the give formulation and discussions in paper do not make sense.
> >
> > **Q5: It is easy to see that Infinite Q_total is not the right one for any discounted MDP.**
> >
> > According to the response, I guess that you may consider TD-learning together with GEM, but it is not mentioned in the paper.
> >
> > **Q6: we generate alpha because we want to learn the explicit credit assignment. Proposition 2 is proved under single-state setting, where there is no need to recover observation, so the reconstruction loss in Equation (10) is removed.**
> >
> > I am asking the motivation to introduce a VAE for learning $\alpha$. The response answers nothing about my question.
> >
> > It seems that the authors also agree Proposition 2 cannot motivate the necessity of reconstruction loss of VAE.

---

> > > ### Author Response · Authors · 2021-11-26
> > > **Thanks for your valuable and insightful comments.**
> > >
> > > **Q1: The IGM condition in QTRAN is maintained by an additional loss to the TD-learning. Do you include that loss function into your method? It requires training a fully centralized value network.**
> > > Please note which sentence we cited QTRAN and QPLEX: "First, it is a sufficient and necessary condition for IGM **under some conditions** as demonstrated by Equation (3), so it has good fitting ability theoretically (Son et al., 2019; Wang et al., 2020b)". We never say that our method can maintaine the sufficient and necessary conditions for IGM, but it can certainly satisfy the sufficient condition of IGM because $\alpha_i$ is a positive value.
> > >
> > > **Q2: In addition, I think the authors do not understand what QPLEX did. The theoretical property of QPLEX is given by its special mixing network rather than the VDN-like sum-form factorization.**
> > > We just say that if the mixing network can maintaine the sufficient and necessary condition for IGM, it will have good fitting ability theoretically as shown by QPLEX. We certainly understand what QPLEX did.
> > >
> > > **Q3: I cannot see what you want to "make sure" through this (i.e., Eq.(4)) formulation. Q_joint is defined as the ground-truth unknown value. How is related to the IGM property? It seems that the authors either do not understand or try to confuse the definition of IGM.**
> > > As shown by the Theorem 1 in the original QTRAN paper, the IGM condition (we mean both sufficient and necessary condition) is defined using the  ground-truth unknown value. Specifically, if the approximated $Q_{total} = \Sigma Q_i + V_{joint}$ is equal to the ground-truth unknown $Q_{joint}$ when the individual actions are jointly optimal (i.e., Eq.(4a) in the original QTRAN), and $Q_{total}$ is larger than $Q_{joint}$  when the individual actions are not jointly optimal (i.e., Eq.(4b) in the original QTRAN), it will satisfy the IGM condition. **Can the reviewer share your understanding about the IGM condition?**
> > >
> > > **Q4: I know you consider other loss functions to train. My concern is that the give formulation and discussions in paper do not make sense.**
> > > Thanks for your suggestion. But we do not see how one-hot vector for alpha can make sense. It means all other agents (but the one has the largest $Q_i$) do not make any contribution to the system? Besides, one-hot vector for alpha is also a special case of our setting $\Sigma \alpha_i=1$.
> > >
> > > **Q5-1: I do not understand why we need to introduce GEM. A trivial maximization solution of Eq.(5) seems to be Qi≡∞.
> > > A5-1: Infinite Q_i will results in infinite Q_total (due to monotonic relation between Qi and Q_total). It is easy to see that infinite Q_total is not the right one for any discounted MDP.
> > > Q5-2: According to the response, I guess that you may consider TD-learning together with GEM, but it is not mentioned in the paper.**
> > > A5-2: We have mentioned the TD-learning in section 4.4. Please note that equation (5) is the explicit credit assignment criterion, and it does not mean we only consider the maximization of Q_total. We also need to make sure the Bellman optimality of Q_total by TD-learning as mentioned in section 4.4.
> > >
> > > **Q6: I am asking the motivation to introduce a VAE for learning . The response answers nothing about my question.**
> > > We are sorry to misunderstand your meaning. The motivation is that the VAE is one of the generative models, and the encoder can generate some alpha, while the decoder loss (i.e., L2-loss to recovery observation) can make sure that the generated alpha is state/observation-dependent so as to better handle multi-state problems. Besides, the KL-loss in VAE can make sure that alphas from different agents are consistent with training goes on. We have mentioned these motivations in our paper (please see the first sentence when introducing the VAE and the first sentence under equation (9)).
> > >
> > > **Q7: It seems that the authors also agree Proposition 2 cannot motivate the necessity of reconstruction loss of VAE.**
> > > We never say this.Proposition 2 is originally written as **"Proposition 2. Under single-state setting, Equation (10) has the same effect as Equation (6)."** We have emphasised that it is under single-state setting. Yes, reconstruction loss of VAE is unnecessary under single-state setting. But reconstruction loss is very necessary to handle the real-world problems consisting of multiple states, since we must make sure that the alpha is state-dependent, and the reconstruction loss between true observation and the recovered observation can be one of the methods (other methods can also be possiable).

---

> > > > ### Comment · Reviewer_wZYm · 2021-11-27
> > > > **I am not convinced by the response**
> > > >
> > > > Thanks for your efforts to clarify these questions. However, I still have lots of concerns on your response.
> > > >
> > > > **Q1: We never say that our method can maintain the sufficient and necessary conditions for IGM, but it can certainly satisfy the sufficient condition of IGM because is a positive value.**
> > > >
> > > > The authors have realized that Eq.(4) does not satisfy the same theoretical properties as QTRAN and QPLEX. The citations here are misleading. You should clarify this difference in the next revision.
> > > >
> > > > **Q2-1: It will have good fitting ability theoretically as shown by QPLEX.**
> > > >
> > > > I am curious about which theory statement in QPLEX implies this conclusion. Please correct me if I misunderstand anything in QPLEX.
> > > >
> > > > **Q2-2: Additional comments**
> > > >
> > > > Note that the value $\alpha_i(\tau_i)Q_i(\tau_i,a_i;\theta_i)$ can be abstracted to a single value $\hat Q_i(\tau_i,a_i;\theta_i)$ since the input of $Q_i$ covers that of $\alpha_i$. Since CTDE does not put constraints on the individual utility $Q_i$, it is equivalent to represent utilities by $\alpha_iQ_i$ and $\hat Q_i$. I do not think $\alpha$ here can indicate credit assignment. There are infinite combinations of $\alpha\times Q_i$ can represent the same value function.
> > > >
> > > > **Q3-1: As shown by the Theorem 1 in the original QTRAN paper, the IGM condition (we mean both sufficient and necessary condition) is defined using the ground-truth unknown value.**
> > > >
> > > > Please check Figure 1 of QTRAN. $Q_{\text{joint}}$ is the output of a network.
> > > >
> > > > In addition, in section 2.3 of QTRAN, the same notation $Q_{\text{joint}}$ refers to the outputs of mixing network.
> > > >
> > > > I do not find any sentences in QTRAN claim $Q_{\text{joint}}$ is the unknown value.
> > > >
> > > > **Q3-2: i.e., Eq.(4b) in the original QTRAN**
> > > >
> > > > Again, $V_{joint}(\tau)$ in Eq.(4b) is implemented by a network.
> > > >
> > > > This paper does not use the same architecture of QTRAN. Why do you claim their theoretical property?
> > > >
> > > > **Q4: It means all other agents do not make any contribution to the system?**
> > > >
> > > > Yes, this case does not make sense but that is my concern.
> > > >
> > > > One-hot vector is a trivial maximization of your objective, so I cannot capture your motivation to introduce that formulation.
> > > >
> > > > **Q5: about GEM**
> > > >
> > > > I cannot understand your logic in response. I suggest authors to further refine the presentation in the next revision.
> > > >
> > > > There are too many unclear concepts presented in this paper. It shows a large gap between the story and implementation.
> > > >
> > > > **Q6: make sure that the generated alpha is state/observation-dependent**
> > > >
> > > > An encoder is sufficient to make $\alpha$ be state-dependent. Why do you introduce a decoder?
> > > >
> > > > The reconstruction loss means different states lead to strictly different credit assignments. It does not make sense.
> > > >
> > > > Again, I would like to emphasize the question in Q2-2, I am not convinced $\alpha$ is a credit assignment here.
> > > >
> > > > **summary**
> > > >
> > > > My biggest concerns are Q2-2 and Q6. I won't increase my score until both these questions are addressed.

---

> > > > > ### Author Response · Authors · 2021-11-28
> > > > > **Thanks for your valuable and insightful comments.**
> > > > >
> > > > > **Q2-2: I am not convinced $\alpha$  is a credit assignment here.**
> > > > > In fact, $\alpha$ and the Transformation Network play the same role as the mixing network of previous methods (e.g., QMIX). If the mixing network of previous methods can do (implicit) credit assignment (this is mentioned in many previous studies as introduced in the Related Work section), why $\alpha$ cannot do this?
> > > > >
> > > > > **Q6: The reconstruction loss means different states lead to strictly different credit assignments. It does not make sense.**
> > > > > In different states, agents take different actions, and the resulting credit assignments should be different (although similar states may have similar credit assignments). We think it makes sense.
> > > > >
> > > > > **Q2/Q3: Comments on IGM**
> > > > > Please note the two terms $Q_{joint}(s,a)$ and $Q_{joint}(s,a; \theta)$ used in QTRAN have different meanings. In QTRAN, the IGM condition is derived using the true (but unknown) $Q_{joint}(s,a)$ and $V_{joint}(s)$. For the neural network implementation, QTRAN uses $Q_{joint}(s,a; \theta)$ to approximate the true (but unknown) $Q_{joint}(s,a)$. In our paper, we use $Q_{total}(s,a; \theta)$ to approximate the true (but unknown)  $Q_{joint}(s,a)$, so as to make the two terms distingshiable. In this perspective, our $Q_{total}(s,a; \theta)$ has the same meaning as QTRAN's $Q_{joint}(s,a; \theta)$.
> > > > >
> > > > >
> > > > > **Overall, we thank the reviewer for the valuable and insightful comments. We will improve our paper in the next version.**

---

> > > > > > ### Comment · Reviewer_wZYm · 2021-11-29
> > > > > > **The response does not answer my question**
> > > > > >
> > > > > > The response does not answer my question. I clearly keep my vote for rejection.
> > > > > >
> > > > > > **Q2-2: If the mixing network of previous methods can do (implicit) credit assignment, why  cannot do this?**
> > > > > >
> > > > > > Yes, the mixing network can be regarded as an implicit credit assignment, but it is not the case here.
> > > > > >
> > > > > > In your formulation, the credit assignment are done jointly by $\alpha_i(\tau_i)$ and $Q_i(\tau_i, a_i)$ since they share the same input $\tau_i$. More specifically, the expressiveness of $Q_i$ dominates $\alpha_i$. I do not think a single value $\alpha_i$ can represent credit assignment, since we do not take any constraints on $Q_i$. The authors' response does not address this point.
> > > > > >
> > > > > > For example, $(\alpha_i,Q_i)=(0.5, 1)$ and $(\alpha_i,Q_i)=(0.25, 2)$ correspond to the same individual value $(\alpha Q)_i=0.5$ in value factorization. However, in the formulation considered by this paper, they correspond to different credit assignment. It does not make sense.
> > > > > >
> > > > > > **Q6: In different states, agents take different actions, and the resulting credit assignments should be different (although similar states may have similar credit assignments).**
> > > > > >
> > > > > > Yes, this argument makes sense, but this property is given by the encoder. I am asking why we need a decoder here. The authors do not answer my question.
> > > > > >
> > > > > > The functionality of the decoder seems to be harmful in the view of credit assignment. It restricts that different states should have **strictly** different credit assignments. According to Q2-2, it strengthens my belief that $\alpha$ does not refer to credit assignment.

---

> > > > > > > ### Author Response · Authors · 2021-11-29
> > > > > > > **Thanks for your valuable and insightful comments.**
> > > > > > >
> > > > > > > **Q2-2: I do not think a single value $\alpha_i$ can represent credit assignment, since we do not take any constraints on $Q_i$. The authors' response does not address this point. The authors' response does not address this point. For example,  $(\alpha_i, Q_i)=(0.5,1)$ and $(\alpha_i, Q_i)=(0.25,2)$ correspond to the same individual value $(\alpha Q)_i=0.5$ in value factorization. However, in the formulation considered by this paper, they correspond to different credit assignment. It does not make sense.**
> > > > > > > Take the example the reviewer given as a concrete example, $(\alpha_i, Q_i)=(0.5,1)$ and $(\alpha_i, Q_i)=(0.25,2)$ certainly represent **different** credit assignment $\alpha_i$ with **different** individual value $Q_i$. Specifically, for $(\alpha_i, Q_i)=(0.5,1)$, agent $i$ has a credit of $0.5$ and other agents share another $0.5$, while for $(\alpha_i, Q_i)=(0.25,2)$, agent $i$ has a credit of $0.25$ and other agents share another $0.75$. It is certainly **different** credit assignment.  Note that it is unsuitable to take $(\alpha Q)_i$ as a whole individual value.
> > > > > > >
> > > > > > > **Q6: In different states, agents take different actions, and the resulting credit assignments should be different (although similar states may have similar credit assignments). Yes, this argument makes sense, but this property is given by the encoder. I am asking why we need a decoder here. The authors do not answer my question.**
> > > > > > > Firstly, we use the variational inference technique to inference the suitable $\alpha_i$ as mentioned in the paper. Under variational inference, it is natural to derive a VAE model, an encoder with a decoder.
> > > > > > > Secondly, it makes sense to produce (very) different credit assignments in (very) different states, and **it makes the same sense to produce similar credit assignments in similar states**. The formmer is easy by only using an encoder, but it is hard to guarantee the latter if no decoder restricts the training of the encoder.

---

> > > > > > > > ### Comment · Reviewer_wZYm · 2021-11-30
> > > > > > > > **The author's response does not make sense to me**
> > > > > > > >
> > > > > > > > There seems to be an unfillable gap between the authors and my view. I think this paper has a large gap between its "story" and implemented algorithm.
> > > > > > > >
> > > > > > > > **Q2: On the credit assignment**
> > > > > > > >
> > > > > > > > Consider two case:
> > > > > > > > - Case \#1: $(\alpha_1, Q_1)(s_1)=(0.25, 12)$, $(\alpha_2, Q_2)(s_1)=(0.75, 8)$, and $(\alpha_1, Q_1)(s_2)=(0.75, 8)$, $(\alpha_2, Q_2)(s_2)=(0.25, 12)$
> > > > > > > > - Case \#2: $(\alpha_1, Q_1)(s_1)=(0.75, 4)$, $(\alpha_2, Q_2)(s_1)=(0.25, 24)$, and $(\alpha_1, Q_1)(s_2)=(0.25, 24)$, $(\alpha_2, Q_2)(s_2)=(0.75, 4)$
> > > > > > > >
> > > > > > > > Both Case \#1 and \#2 have $(\alpha_1 Q_1)(s_1)=3$, $(\alpha_2 Q_2)(s_1)=6$, $(\alpha_1 Q_1)(s_2)=6$, $(\alpha_2 Q_2)(s_2)=3$. Since these two solutions are the same under TD-loss and ECA-loss. In addition, due to the symmetry of $\alpha$, both cases are the same under the VAE objective. Thus the objective function used by this paper cannot distinguish these two solutions. Similarly, we can construct infinite combinations of $\alpha_i\times Q_i$ to achieve the same situation.
> > > > > > > >
> > > > > > > > More specifically, for any given $\alpha$, we can simply modify the value of $Q_i$ to get the same global value $Q_{total}$ for TD-learning. It shows that $\alpha$ is just a hidden neuron within value factorization. I cannot see why $\alpha$ refers to any meaningful things such as credit assignment.
> > > > > > > >
> > > > > > > > **Q6: On the VAE loss**
> > > > > > > >
> > > > > > > > Why do we need **very** different credit assignments for **very** different states? In practice, the "state" refers to agents' observations (e.g., given by sensors), which is extremely redundant. It is quite common that different observation sets correspond to the same system configuration. If the underlying system status is the same, we should have the same credit assignment to develop a consistent decision-making system. I definitely do not agree authors' arguments.

---

> > > > > > > > > ### Author Response · Authors · 2021-11-30
> > > > > > > > > **Thanks for your valuable and insightful comments.**
> > > > > > > > >
> > > > > > > > > The reviewer's understanding for the IGM-based credit assignment does not make sense to me (The first author of this paper).
> > > > > > > > >
> > > > > > > > > **Q2-1: On the credit assignment**
> > > > > > > > > The reviewer writes that case #1 $(\alpha_1, Q_1)(s_1)=(0.25, 12)$ and case #1 $(\alpha_1, Q_1)(s_1)=(0.75, 4)$. But the example cannot be true in real environment. Specifically, if the dnn parameters are fixed, how can we get different $\alpha_i$ and $Q_i$ is the same state $s_1$? Unless the dnn parameters are optimized during training. Therefore, there is only one case is a good credit assignment.
> > > > > > > > >
> > > > > > > > >
> > > > > > > > > **Q2-2: More specifically, for any given $\alpha$, we can simply modify the value of $Q_i$ to get the same global value  $Q_{total}$ for TD-learning. It shows that $\alpha$ is just a hidden neuron within value factorization. I cannot see why $\alpha$ refers to any meaningful things such as credit assignment.**
> > > > > > > > > The above argument is also suitable for QMIX and previous methods. In these methods, we can also modify the mixing network and $Q_i$ to get the same global value  $Q_{total}$ for TD-learning. But researchers generally think QMIX can learn good credit assignment. Could the reviewer give some comments on this?
> > > > > > > > >
> > > > > > > > >
> > > > > > > > >  **Q6: On the VAE loss**
> > > > > > > > > The reviewer just catch the half meaning of our sentense. Obvioursly, we just want to express that it makes sense to produce different credit assignments in different states, and it makes the same sense to produce similar credit assignments in similar states (as the reviewer said: "If the underlying system status is the same, we should have the same credit assignment to develop a consistent decision-making system."). We do not think our arguments have any problem.

---

> > > > > > > > > > ### Comment · Reviewer_wZYm · 2021-12-02
> > > > > > > > > > **Some clarifications on my questions**
> > > > > > > > > >
> > > > > > > > > > **Q2: On the credit assignment**
> > > > > > > > > >
> > > > > > > > > > Consider "case #1" and "case #2" as two different solutions (represented by different network parameters). They have the same values of $Q_{total}$ and $\alpha_i*Q_i$, i.e., they equally optimize TD-loss and ECA-loss. They have symmetric $\alpha$, i.e., they equally optimize VAE-loss. An RL algorithm is to determine which one is the policy for agents. Under the objective designed by this paper, these two solutions seem to be the same. Similarly, we can construct infinite combinations of $\alpha_i$ and $Q_i$ to achieve the same condition. It refers to the following argument.
> > > > > > > > > >
> > > > > > > > > > From the view of value factorization, the global value $Q\_{total}$ is first decomposed to $\sum (\alpha\*Q)\_i$ and then to $\sum \alpha\_i*Q\_i$. This paper claims $\alpha$ corresponds to "credits". However, we can simply modify the values of $Q_i$ to adapt to the changes in $\alpha_i$ while keeping the global value $Q_{total}$ does not change. It raises a concern that VAE-loss seems to be no interactions with TD-loss, i.e., your learned "credits" have no impact on the decision-making.
> > > > > > > > > >
> > > > > > > > > > In terms of IGM credit assignment, "credits" refer to the values of $\alpha_i* Q_i$. I do not understand why this paper emphasizes $\alpha$ as credits.
> > > > > > > > > >
> > > > > > > > > > **Q6: On the VAE loss**
> > > > > > > > > >
> > > > > > > > > > I do not think "*produce different credit assignments in different states*" makes sense. The utilization of VAE eliminates the possibility for "*states look very different but have the same underlying system status*" to have the same credit assignment. Do you mean we should not optimize the reconstruction loss too well?
> > > > > > > > > >
> > > > > > > > > > **Suggestions**
> > > > > > > > > >
> > > > > > > > > > If we repeat the experiments in Table 2, can we get similar credit assignments in almost all runs? This experiment can be a good addition to this paper.

---

> > > > > > > > > > > ### Author Response · Authors · 2021-12-02
> > > > > > > > > > > **Thanks for your valuable and insightful comments**
> > > > > > > > > > >
> > > > > > > > > > > **Q2-1: On the credit assignment. However, we can simply modify the values of $Q_i$ to adapt to the changes in $\alpha_i$ while keeping the global value $Q_{total}$ does not change.**
> > > > > > > > > > > I do not think this is a problem. Think that QMIX and many previous methods also have this property: we can modify QMIX's $Q_i$ to adapt to the changes in the values of the mixing network while keeping the global value $Q_{total}$ does not change. How does this property matter?
> > > > > > > > > > >
> > > > > > > > > > > **Q2-2: On the credit assignment. It raises a concern that VAE-loss seems to be no interactions with TD-loss, i.e., your learned "credits" have no impact on the decision-making.**
> > > > > > > > > > > The VAE-loss has interactions with the TD-loss. Specifically, 1) the VAE-loss updates $\alpha_i$, which influences the $Q_{total}$, and further influences the TD-loss; 2) in turn, the TD-loss will train the parameters that generate $\alpha_i$ (as well as $Q_i$), which will influence the value of VAE-loss.
> > > > > > > > > > >
> > > > > > > > > > > **Q2-3: On the credit assignment. In terms of IGM credit assignment, "credits" refer to the values of $\alpha_i * Q_i$. I do not understand why this paper emphasizes $\alpha$ as credits.**
> > > > > > > > > > > This is a very fundamental comment. It represents the cognitive gap between the reviewer and me (the first author of this paper). **Could the reviewer give me the papers/citations that claim " 'credits' refer to the values of $\alpha_i * Q_i$"?**  We indeed wrote in our paper that "Second, it allows us to intuitively interpret the weight $\alpha_i(\tau_i)$ as the importance of $Q_i$, which can be analyzed to understand the concrete credit assignment (please see the experiments)."
> > > > > > > > > > >
> > > > > > > > > > > **Q6: On the VAE loss. I do not think "produce different credit assignments in different states" makes sense. The utilization of VAE eliminates the possibility for "states look very different but have the same underlying system status" to have the same credit assignment. Do you mean we should not optimize the reconstruction loss too well?**
> > > > > > > > > > > We never say "we should not optimize the reconstruction loss too well". Firstly, we do not think distinguishing the "states look very different but have the same underlying system status" and "states look very different and their underlying system states are truely very different" is the main topic of our paper. Secondly, the former situation is more common or the latter situation is more common? Thirdly, (we remind the reviewer that) the reviewer should also disagree our claim that "produce similar credit assignments in similar states", since there is also a probability that "the states look very similar but have very different underlying system states".
> > > > > > > > > > >
> > > > > > > > > > > **If we repeat the experiments in Table 2, can we get similar credit assignments in almost all runs? This experiment can be a good addition to this paper.**
> > > > > > > > > > > In fact, the results in Table 2 can be reproduced in almost all runs! But we do find a few outlines (in many many runs) due to the randomness introduced by exploration.
> > > > > > > > > > >
> > > > > > > > > > >
> > > > > > > > > > > **We thank reviewer wZYm for her efforts to improve our paper. When the anonymity period is over, we are glad that reviewer wZYm could contact me (the first author of this paper) for further discussion.**

---

### Official Review · Reviewer_49nJ · 2021-11-01

**Correctness:** 3
**Technical Novelty And Significance:** 3
**Empirical Novelty And Significance:** 2
**Recommendation:** 5
**Confidence:** 3

**Main Review:**

Overall, the paper is interesting to read, and it advances the state-of-the-art multi-agent RL literature by introducing a new MARL method that considers a novel form of credit assignment. That said, I have several concerts about the current results and their exposition, listed below together with the strengths of the paper.

**Strengths:**

- *The paper studies an important problem.* It explores one of the fundamental challenges in multi-agent RL, the credit assignment problem, and it contributes novel ideas relevant for making cooperative multi-agent RL more effective. It studies an explicit approach to the multi-agent credit assignment problem, in which agents estimate their contributions to the joint performance.

- *The novel algorithmic approach outperforms baselines in standard multi-agent RL testbeds.* Based on the presented results, the proposed method, ECAQ, seems to outperform baselines from prior work. More specifically, ECAQ's performance is either comparable to or better than recent MARL methods. The proposed MARL design (NN architecture in Figure 1) seems novel.

- *Related work is covered well and the paper explains how its main algorithmic contribution is grounded in theory.* The paper extensively surveys the prior work on the credit assignment problem in  MARL, providing satisfactory explanations for the most important references. It also develops the proposed MARL method by reasoning about the exact solutions to the *explicit* credit assignment problem for simple domains.

**Weaknesses:**

- *The contributions of the paper are somewhat incremental, while the paper could be considerably improved in terms of presentation.* It is hard to disentangle the core novel ideas that the paper introduces from existing works. While I believe I understood  the main contributions of the paper, I generally found the details hard to follow. In my opinion, the paper is not written very well and it contains quite a few typos.

- *Theoretical results are not rigorous enough.* For example, the proof of Proposition 1 is actually not a formal proof. This is best demonstrated by the sentence: *...it is easy to prove that $Q_{total}$ will also be convex given a specific weighting vector;...*. Generally, some of the theoretical claims look quite informal and/or are not rigorously stated. For example,  in Proposition 2, what does *the same effect* mean? Some of the modeling assumptions are not clearly explained. For example, in Eq. 7, we have that policy parameters $\theta_i$ are updated based on the policy parameters of other agents, i.e., $\theta_j$, but this update rule does not seem to be justified in the text.

- *Experimental results are not clearly explained and look somewhat incomplete.* Regarding the experiments, I've found the main results interesting, but this whole section could be considerably improved. a) There are quite a few formatting issues. For example, the exact meaning of the bold numbers in Table 1 does not seem to be explained, or in Figure 2, x and y axis are barely visible.  b) It is not clear what the baselines selection procedure is for different environments. For example, in Figure 3, the results are obtained for a cooperative environment from Lowe et al. 2017. How does the approach from Lowe et al. 2017 perform compared to the proposed approach? Why are some baselines (QPLEX, WQMIX, LICA, etc.) missing in Figure 3?  c) Some of the  paragraphs in the experimental section are ambiguous. For example, the game from the last paragraph in Section 5.2 is taken from Son et al. 2019. However, the 3 settings mentioned in this paragraph are not clearly explained; perhaps explaining how these results relate to the ones in Son et al 2019 could help. It is also not clear why these results show *the good credit assignment ability* of ECAQ, especially since the discussion is not focused on comparing ECAQ to other approaches.


**Summary Of The Paper:**

The paper studies the credit assignment problem in multi-agent reinforcement learning (MARL), focusing on the centralized training with decentralized executing paradigm. The paper introduces a new MARL method based on value function factorization, and it experimental showcases its benefits over existing approaches. This new method aims to more explicitly account for agents' contributions to the joint performance, and the paper aims to justify its components using a simple theoretical analysis. The core evaluation is experimental, and it is based on standard MARL experimental testbeds from prior work.

**Summary Of The Review:**

The paper contains interesting contributions, however its theoretical and experimental results could be improved: formal claims should be more rigorously stated, while experimental results could be expanded. The presentation of the paper could be considerably improved.

---

> ### Author Response · Authors · 2021-11-12
> **Thanks for your valuable and insightful comments.**
>
> Thanks for your valuable and insightful comments. Below we will address your concerns in detail.
>
> **Question**: while the paper could be considerably improved in terms of presentation.
> **Answer**: Thanks for your suggestion and effort. We will present more details about the technique and revise the typos to make the paper easier to follow in the next version.
>
> **Question**: Theoretical results are not rigorous enough.
> **Answer**: Due to space limitation, we only provide a proof skeleton. We will present more details about the technique in the next version. Thanks for your suggestion and effort.
>
> **Question**: Experimental results are not clearly explained and look somewhat incomplete.
> **Answer**: Thanks for your suggestion and effort. A) We will revise the formatting issues about Table 1 and Figure 2 in the next version. B) The Cooperative Navigation environment has discrete action space, so we do not compare MADDPG (which is a method for continuous action space); we have compared QPLEX, WQMIX, LICA in Cooperative Navigation, but they work worse than VDN and ECAQ; due to space limitation, we do not shown those results. C) The matrix game is the same as QTRAN and please note that the experiments are mainly used to test whether ECAQ has learned explainable credit assignment, not focus on comparing ECAQ to other approaches.

---

> > ### Comment · Reviewer_49nJ · 2021-11-29
> > **I keep my score**
> >
> > Thanks for providing the clarifications. Based on the overall discussion, I believe that the paper could be considerably improved, so I will keep my score as it is.

---

> > > ### Author Response · Authors · 2021-11-30
> > > **Thanks**
> > >
> > > We thank Reviewer 49nJ for the efforts and valuable time to improve our paper. We do see the discussion in a positive light. Thanks again.

---

### Official Review · Reviewer_56iQ · 2021-11-03

**Correctness:** 4
**Technical Novelty And Significance:** 3
**Empirical Novelty And Significance:** 3
**Recommendation:** 6
**Confidence:** 4

**Main Review:**

Overall speaking, I think this paper is written clearly and easy to follow, and I think it is quite valuable to the research area of Multi-agent Cooperative Learning.

The authors introduced a simple but very helpful idea of "explicit credit assignment" into existing IGM-based joint Q-learning studies, by using maximizing Q_{total} as a criterion for realizing efficient credit assignment. To the best of my knowledge the proposed technique is novel, and the empirical experiment evaluation results appear to be convincing and demonstrate that the proposed method achieve superior performance compared to several advanced SOTA baselines (e.g., WQMIX, DOP).

On the other hand, although the current experimental results are already somewhat exciting, it would be better and even more convincing if we could see it scaled up (to the extent possible) in the other envs, to make sure the NNs don't break anything at scale. Such scale testing of the proposed approach would be more valuable to understand better the power and limits of the proposed novel method.

Moreover, for better clarity and fair comparison as well as better replicatibility,  it would be good to for the authors to explicitly clarify in the paper how much tuning efforts (e.g., tune the architecture) were spent there to get these results, for the proposed ECAQ method as well as other baseline methods.





**Summary Of The Paper:**

In this paper, the authors proposes a novel scheme named ECAQ for learning an explicit credit-assignment scheme for CTDE tasks, which in its theoretical form should converge to the optimal solution and also empirically evaluates the DNN form. The main idea is to use maximizing Q_{total} as a criterion for realizing efficient credit assignment for  for multi-agent joint Q-learning. The authors conducted experiments to demonstrate that the proposed ECAQ technique achieves interpretable credit assignment and superior performance compared to several advanced baselines.

**Summary Of The Review:**

Briefly speaking, I think the paper is well-written and easy to follow, and he results seems to be technically solid based on my check. The ideas and contents in the paper seems to be interesting and novel, and the experimental results are also providing some reasonable justification of the values of the proposed new approach.

There are some minor areas that might be improved about the empirical experiment evaluations part, as described in the main review section, but generally speaking, I think the paper is valuable to the research community of  of Multi-agent Cooperative Learning, and I don't have too much concern about accepting the paper for publication on ICLR.

---

> ### Author Response · Authors · 2021-11-12
> **Thanks for your valuable and insightful comments.**
>
> Thanks for your valuable and insightful comments. Below we will address your concerns in detail.
>
> **Question**: it would be better and even more convincing if we could see it scaled up (to the extent possible) in the other envs, to make sure the NNs don't break anything at scale. Such scale testing of the proposed approach would be more valuable to understand better the power and limits of the proposed novel method.
> **Answer**: Thanks for your suggestion and effort. Currently, we test our methods on MMM2 map, which has 10 controllable agents and 12 internal enemies, and 10-vs-10 Cooperative Navigation. We will try environments with more agents in the future.
>
> **Question**: for better clarity and fair comparison as well as better replicatibility, it would be good to for the authors to explicitly clarify in the paper how much tuning efforts (e.g., tune the architecture) were spent there to get these results, for the proposed ECAQ method as well as other baseline methods.
> **Answer**: We did not spend much effort on fine-tuning the architecture or the hyperparameters of ECAQ. Specially, PicaQ is implemented based on the popular PyMARL framework with the default configurations/hyperparameters. This is because we aim at demonstrating that explicit credit assignment is useful for the IGM-based joint Q-learning, rather than tuning the architecture or the hyperparameters to beat all baseline methods.

---

> > ### Comment · Reviewer_56iQ · 2021-11-29
> > **I keep my score**
> >
> > Thanks for providing the helpful clarifications.My questions have been addressed, but based on the overall discussions including all reviewers and authors, it does seem that the paper could still be considerably improved. I'm now at the borderline of 5 or 6, just slightly leaning towards keeping my original score at 6. I won't push for acceptance if other reviewers have strong opinions for objection.

---

> > > ### Author Response · Authors · 2021-11-30
> > > **Thanks**
> > >
> > > We thank Reviewer 56iQ for the efforts and valuable time to improve our paper. We do see the discussion in a positive light. Thanks again.

---

### Decision · Program_Chairs · 2022-01-20

**Decision:**

Reject

**Comment:**

This paper provides an interesting method to address the CTDE problem in MARL. While the experiments are promising, the theory is either insufficient or not rigorous. One of the reviewers believe that there is a flaw in the paper. There was an extensive discussion among the authors and the reviewer. The authors could not convince the reviewer for the apparent flaw.